# LLM-Guided Communication for Cooperative Multi-Agent Reinforcement Learning

**Sangjun Bae** [1] **Yisak Park** [1] **Sanghyeon Lee** [1] **Seungyul Han** [1*]

## Abstract

Communication is a key component in multi-agent reinforcement learning (MARL) for mitigating partial observability, yet prior approaches often rely on inefficient information exchange or fail to transmit sufficient state information. To address this, we propose LLM-driven Multi-Agent Communication (LMAC), which leverages an LLM's reasoning capability to design a communication protocol that enables all agents to reconstruct the underlying state as accurately and uniformly as possible. LMAC iteratively refines the protocol using an explicit state-awareness criterion, improving state recovery while narrowing differences in agents' knowledge. Experiments on diverse MARL benchmarks show that LMAC improves state reconstruction across agents and yields substantial performance gains over prior communication baselines. Code is available at https://saaangjun.github.io/LMAC/.

## 1. Introduction

Cooperative multi-agent reinforcement learning (MARL) has emerged as a key paradigm for solving complex tasks that require multi-agent collaboration, including autonomous driving, network management, and strategic simulation games (Nguyen et al., 2020; Orr & Dutta, 2023; Chen et al., 2023a). To scale MARL toward such real-world applications, a wide range of research directions have been actively pursued, including exploration (Jo et al., 2024; Park et al., 2026), robustness (Lee et al., 2025; 2026), and communication (Zhu et al., 2024; Jo et al., 2026). A core challenge commonly addressed across these directions is partial observability, and to mitigate it, the centralized training with decentralized execution (CTDE) framework (Lowe et al.,

2017) has been widely adopted, exploiting global state information during training while relying on local observations at execution time. However, agents can still face partial observability under decentralized execution, and learning may fail on tasks where explicit communication is necessary.

To address partial observability more directly, a range of communication methods that enable agents to exchange messages has been actively studied in MARL. These methods differ in how messages are structured, routed, and integrated into each agent's policy to support coordination. Early work primarily considered broadcast communication, where the same message is sent to all agents (Sukhbaatar et al., 2016; Guan et al., 2022). To enable more efficient communication, subsequent methods allow agents to exchange local messages directly with one another (Das et al., 2019; Wang et al., 2019; Yuan et al., 2022). Representative approaches include TarMAC, which uses attention to weight received messages (Das et al., 2019), SMS, which scores message contribution using Shapley values (Xue et al., 2022), and T2MAC, which integrates messages using evidence-based fusion (Sun et al., 2024). However, these methods do not explicitly address how efficiently and accurately agents can reconstruct state information, which can result in messages that do not sufficiently capture the underlying state.

Efficient state reconstruction requires exchanging only the essential messages needed for recovery, yet identifying them is challenging because it demands understanding the task objective and the relationship between the state and observations. To address this challenge, we propose **LLM-driven Multi-Agent Communication (LMAC)**, which leverages a large language model (LLM)'s reasoning capability to design a communication protocol by describing the task goal and each dimension of the state and observations in natural language. LMAC builds on Reflexion (Shinn et al., 2023) to generate an initial protocol that encourages efficient communication, then iteratively expands it by adding only the elements needed for state reconstruction based on feedback. As shown in Fig. 1, refinement follows two objectives: *recovery enhancement*, which improves recovery when agents fail to reconstruct the state, and *imbalance mitigation*, which supplements messages for agents with insufficient information. Guided by criteria derived from collected transition

---

[1]Graduate School of Artificial Intelligence, UNIST, Ulsan, South Korea. Correspondence to: Seungyul Han <syhan@unist.ac.kr>.

*Proceedings of the 43rd International Conference on Machine Learning*, Seoul, South Korea. PMLR 306, 2026. Copyright 2026 by the author(s).

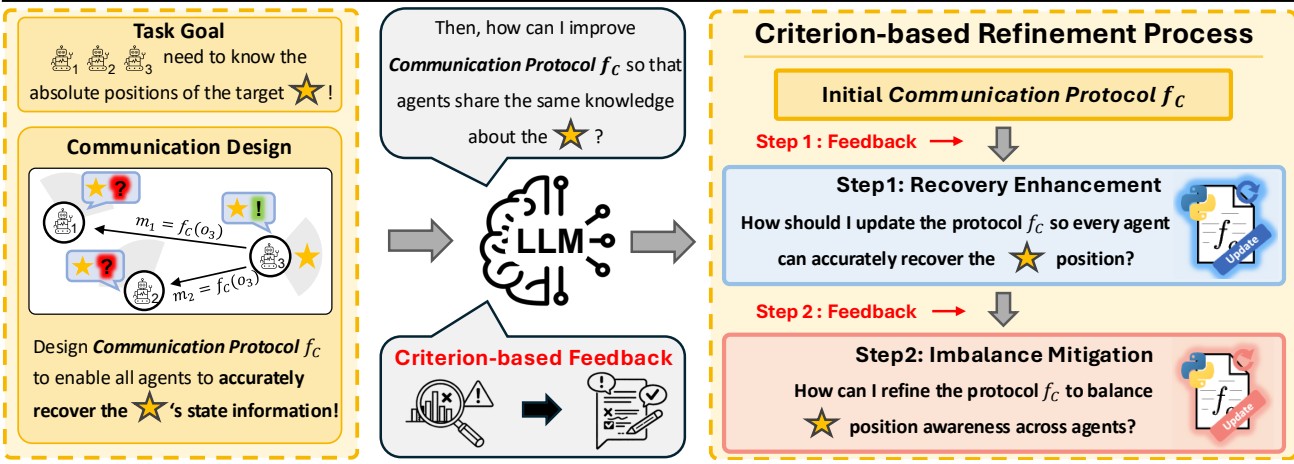

*Figure 1.* Protocol refinement in LMAC. Given natural-language descriptions of the task goal and the state and observation dimensions, the LLM designs an initial communication protocol to support accurate state reconstruction (star position in the figure). The protocol is then refined via criterion-based two-step feedback to make reconstruction more accurate and uniform by sharing missing local information: Step 1 improves each agent's state reconstruction accuracy, and Step 2 reduces information gaps across agents.

data, LMAC adds only messages required for recovery. Experiments show that LMAC improves state reconstruction and performance over prior communication methods on diverse MARL benchmarks that require communication.

## 2. Related Works

**Communication in MARL.** A broad line of work has studied communication to mitigate partial observability in MARL. Broadcast-based approaches include early designs such as hidden-state averaging and bidirectional chains (Sukhbaatar et al., 2016; Foerster et al., 2016), as well as more centralized variants with dedicated encoders or global coaches (Guan et al., 2022; Shao et al., 2023). In contrast, agent-wise communication methods enable more selective exchange, for example via soft attention and query-based matching (Das et al., 2019; Sun et al., 2024), or by learning graph-structured communication to capture relational dependencies (Niu et al., 2021; Hu et al., 2024). Several works further introduce gating mechanisms and regularizers to optimize what, when, and with whom to communicate (Singh et al., 2019; Wang et al., 2019; Ding et al., 2020; Zhang et al., 2020; Xue et al., 2022; Yuan et al., 2022).

**LLMs for Reasoning.** LLMs (Vaswani et al., 2017) trained on massive data, such as GPT (OpenAI, 2023), Gemini (Gemini Team & Google, 2023), and Claude (Anthropic, 2024), have been widely adopted for solving complex reasoning tasks. Chain-of-thought (CoT) prompting (Wei et al., 2022) has been extended to more structured reasoning formats (Zhou et al., 2022; Yao et al., 2023; Besta et al., 2024; Sel et al., 2023; Yang et al., 2024). Zero-shot reasoning (Kojima et al., 2022) and instruction tuning with self-generated data (Wang et al., 2023) further highlight the versatility of prompting. ReAct (Yao et al., 2022) combines reasoning traces with environment interactions, while iterative

refinement methods such as Reflexion (Shinn et al., 2023), Retroformer (Chen et al., 2024), and Expel (Zhao et al., 2024) enable repeated self-correction.

**LLMs for RL.** LLMs have also been used to improve RL across diverse roles, including reward design (Nair et al., 2022; Adeniji et al., 2023; Chu et al., 2023; Ma et al., 2024; Xie et al., 2023), trajectory summarization and task transformation (Du et al., 2023; Yuan et al., 2023; Qiu et al., 2024), and state representation (Chen et al., 2023b; Wang et al., 2024; Da et al., 2024). They have additionally been employed as policy networks (Li et al., 2022; Zitkovich et al., 2023; Shi et al., 2023) or for grounding actions and coordination policies (Ahn et al., 2022; Hu & Sadigh, 2023). More recently, LLMs have been explored in MARL to strengthen multi-agent coordination (Li et al., 2025; Agashe et al., 2025) and to perform language-based communication by mimicking conversations among LLM agents in simple environments (Li et al., 2024).

In this paper, we focus on improving communication protocols in MARL using LLM reasoning. Prior work (Li et al., 2024) requires online interaction between LLM agents and the environment at each timestep, which can increase LLM usage cost and is typically most suitable when the environment provides a text-world interface. In contrast, our approach forms language feedback using criteria constructed solely from RL transition data, without online LLM interaction. This design substantially reduces LLM usage cost, provides explicit criteria that target improved state recovery, and remains applicable to general multi-agent environments given task instructions.

## 3. Background

**Dec-POMDPs with Communication.** Cooperative MARL with communication can be formalized as a

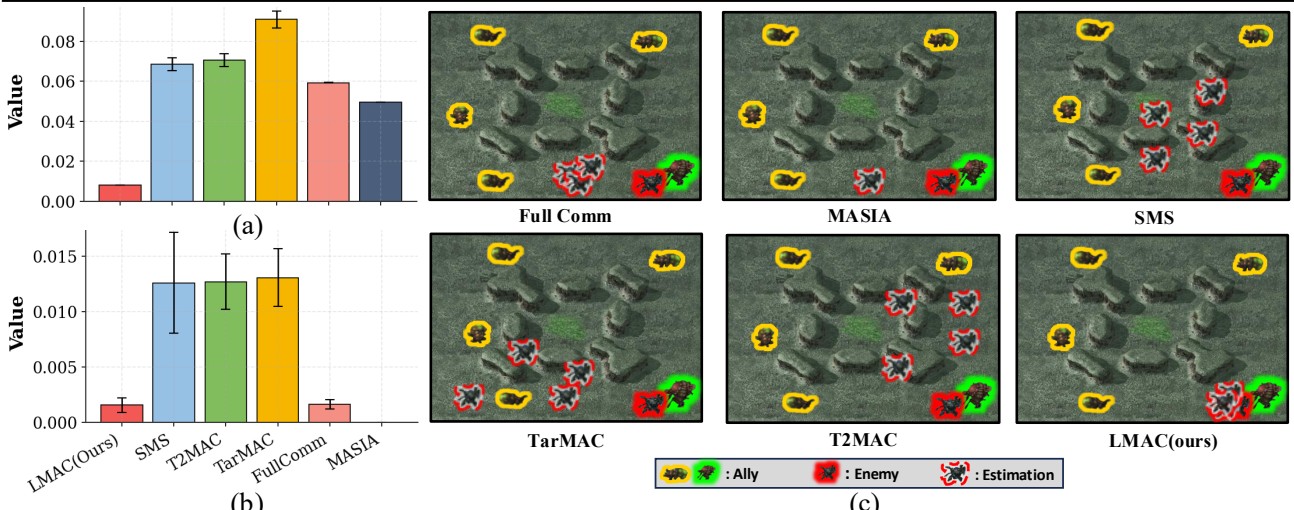

*Figure 2.* Comparison on the StarCraft II environment after 2M timesteps. (a) Average reconstruction error of the enemy position. (b) Inter-agent variance of the reconstruction error. (c) The true positions of ally agents (green/yellow) and the enemy (red), and each ally's estimated enemy position. For all methods, the metrics are computed by performing an additional state-inference step.

decentralized partially observable Markov decision process with communication (Comm-Dec-POMDP), $G = \langle S, A, P, R, O, \mathcal{O}, I, n, \gamma, \mathcal{M} \rangle$. Here, $S$ is the global state space, $A$ the joint action space, $P$ the transition dynamics, $R$ the reward function, $O$ the observation function with observation space $\mathcal{O}$, $I$ the set of $n$ agents, $\gamma$ the discount factor, and $\mathcal{M}$ the message space. At timestep $t$, agent $i$ receives an observation $o_t^i = O_i(s_t)$ and selects an action $a_t^i$ via a decentralized policy $\pi^i(\cdot \mid \tau_t^i)$ conditioned on its trajectory $\tau_t^i := (o_0^i, a_0^i, \ldots, o_t^i)$. The objective is to maximize the expected cumulative reward $\mathbb{E}\left[\sum_{t=0}^{T-1} r_t\right]$. Under the CTDE paradigm, training typically learns a global action-value function $Q_{\text{tot}}(s_t, \tau_t, \mathbf{a}_t)$ using the global state $s_t$, together with individual utilities $Q^i(\tau_t^i, a_t^i)$, enabling credit assignment for selecting high-value joint actions.

When incorporating communication under CTDE to mitigate partial observability, agents are equipped with a mechanism to exchange messages $m_t^i \in \mathcal{M}$ at each timestep, yielding utilities and policies of the form $Q^i(\tau_t^i, m_t^i, a_t^i)$ and $\pi^i(\cdot \mid \tau_t^i, m_t^i)$. In broadcast communication, the same message is shared across agents, whereas in agent-wise communication, messages can differ by agent.

**LLM Reasoning Formulation.** The CoT method can be formulated as follows: Given an input prompt $x$, an LLM first produces reasoning tokens $z \sim f_\theta^{\text{LLM}}(x)$ and then generates the final output $y \sim f_\theta^{\text{LLM}}(x, z)$, where $f_\theta^{\text{LLM}}$ denotes the LLM with parameters $\theta$. Among various approaches for improving reasoning, feedback-driven iterative refinement is particularly relevant to our setting. Reflexion (Shinn et al., 2023) updates reasoning across iterations as follows: At iteration step $k$, the model outputs $y^{(k)} \sim f_\theta^{\text{LLM}}(x, z^{(k)})$, derives a feedback sentence $c^{(k+1)} \sim f_\theta^{\text{LLM}}(x, \tilde{x}, z^{(k)}, y^{(k)})$, where $\tilde{x}$ is a feedback instruction, and refines the reason-

ing via $z^{(k+1)} \sim f_\theta^{\text{LLM}}(x, c^{(k+1)})$, leading to $y^{(k+1)} \sim f_\theta^{\text{LLM}}(x, z^{(k+1)})$. In our formulation, $x$ corresponds to the task instruction, $y^{(k)}$ is the communication protocol produced at step $k$, and $c^{(k)}$ provides criterion-based language feedback aligned with the two-step objectives in Fig. 1.

## 4. Methodology

### 4.1. Motivation: Agent-wise Communication Design for State Information Recovery

To mitigate partial observability, broadcast methods such as MASIA (Guan et al., 2022) share common information across agents by learning a state-reconstructing latent representation, while agent-wise methods aim to reduce communication cost, including FullComm, which shares other agents' observations for individual state reconstruction, and message-prioritization approaches such as TarMAC (Das et al., 2019), SMS (Xue et al., 2022), and T2MAC (Sun et al., 2024) introduced in Section 1. However, despite these developments, existing methods can still leave inefficiencies in recovering state information.

Fig. 2 illustrates this issue on a StarCraft II task where allies (green/yellow) must infer an enemy (red) position and attack simultaneously. The Overseer (green) directly observes the enemy and could share this key information to help reconstruct the enemy position. Fig. 2(a) reports the mean reconstruction error of the enemy position from agents' messages at 2M timesteps and shows that even MASIA and FullComm can remain inaccurate due to redundant information. Fig. 2(b) reports the inter-agent variance of reconstruction error, indicating that other agent-wise methods yield non-uniform reconstruction across agents. As a result, Fig. 2(c) compares the ground-truth enemy position with agents' estimates, showing that existing methods still fail to

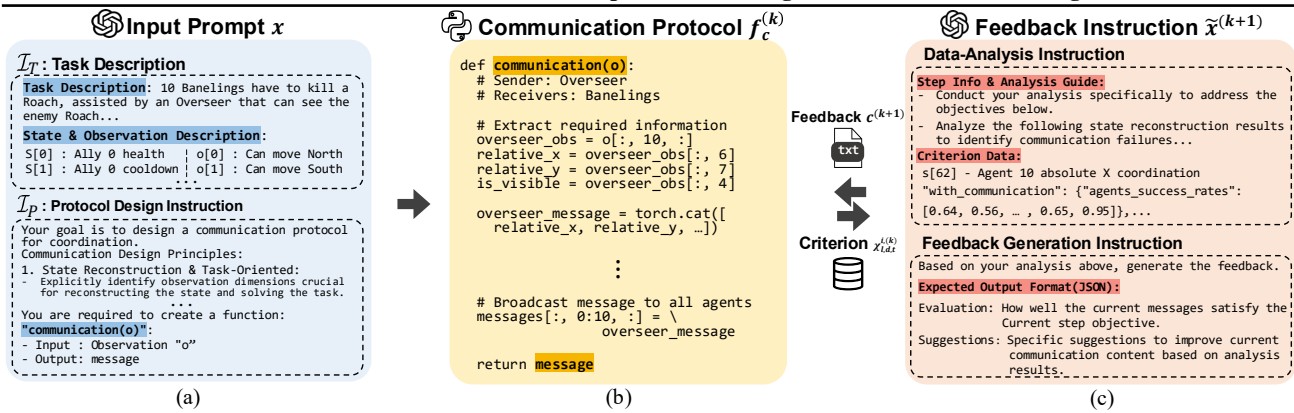

*Figure 3.* Overview of the iterative communication protocol refinement framework: (a) Input prompt $x$ with task description $\mathcal{I}_T$ and design instruction $\mathcal{I}_P$, (b) generated protocol $f_C^{(k)}$ mapping local observations to agent-specific messages, and (c) feedback instruction $\tilde{x}^{(k+1)}$ guiding the next optimization step via step-wise criterion-based analysis and feedback generation.

estimate the enemy position reliably.

These results highlight the need for agent-wise communication protocols that enable accurate and consistent state recovery across agents. Achieving this goal requires identifying which dimensions of each agent's local observations are essential for reconstruction and which are redundant. To this end, we propose the LMAC framework, which uses LLM reasoning to infer what messages each agent should exchange. LMAC first designs an initial protocol that exchanges minimal messages, then improves it via criterion-based language feedback constructed solely from RL transition data, adding only the messages needed to enhance recovery and mitigate inter-agent information imbalance, as described in Fig. 1. With this design, as illustrated in Fig. 2, LMAC achieves lower reconstruction error while maintaining low inter-agent variance, enabling all agents to estimate the enemy position accurately.

### 4.2. LLM-driven Communication Protocol Design

In this section, we design an agent-wise communication protocol to improve state awareness based on Reflexion. As introduced in Section 3, the original Reflexion loop uses a fixed feedback instruction $\tilde{x}$ across iterations. Our protocol design, however, requires iteration-specific objectives. We therefore replace $\tilde{x}$ with step-wise feedback instructions $\tilde{x}^{(k+1)}$ to generate the feedback sentence $c^{(k+1)}$ at each iteration. Our goal is to obtain an executable, code-based communication protocol $f_C^{(k)}$ that maps agents' observation histories $(\tau_t^0, \ldots, \tau_t^{n-1})$ to agent-specific messages $\boldsymbol{m}_t^{(k)} := (m_t^{0,(k)}, \ldots, m_t^{n-1,(k)})$, i.e.,

$$f_C^{(k)} \sim f_\theta^{\text{LLM}}(x, z^{(k)}), \quad k \in \{0, 1, 2\}, \quad (1)$$

where $z^{(k)} \sim f_\theta^{\text{LLM}}(x, c^{(k)})$ denotes the reasoning tokens produced by the Reflexion feedback loop, with $c^{(0)} = \emptyset$. Here, we adopt a code-based protocol so that messages can be produced directly from agents' observations without on-

line LLM interaction. To achieve the objectives illustrated in Fig. 2, we specify the input prompt $x$ and provide criterion-based language feedback at each iteration, as detailed next.

**Input Prompt $x$.** To generate the communication protocol, we design an input prompt $x$ and provide it to the LLM. The prompt consists of a task description $\mathcal{I}_T$ and a protocol design instruction $\mathcal{I}_P$, as summarized in Fig. 3(a). The task description $\mathcal{I}_T$ specifies the cooperative goal and provides natural-language descriptions of each state and observation dimension, guiding the LLM to interpret the task and its information structure. Building on this, the protocol design instruction $\mathcal{I}_P$ specifies the required input–output format so that the protocol maps local observations to messages, and sets the design objective of transmitting only the information essential for other agents to reconstruct the global state.

**Criterion-based Feedback Instruction.** At each iteration, we refine the LLM-generated protocol to improve agents' ability to reconstruct the global state. To quantify agent-wise state awareness and derive criterion-based feedback, we use an offline transition dataset $\mathcal{B}$. Given the protocol at iteration $k$, denoted by $f_C^{(k)}$, we train an auxiliary decoder $D_\phi^{(k)}$ that reconstructs the global state from an agent's local trajectory $\tau_t^i$ and compare reconstructions with and without the message $m_t^{i,(k)}$ produced by $f_C^{(k)}$. The resulting reconstruction error provides a quantitative signal, which we convert into a state-awareness indicator (SAI) by thresholding per-dimension accuracy. For agent $i$, state dimension $d$, and timestep $t$, we define the decoder predictions with ($l = 1$) and without ($l = 0$) the message as

$$\hat{s}_{1,d,t}^i = D_\phi^{(k)}(\tau_t^i, m_t^{i,(k)}, i)\big|_d, \quad \hat{s}_{0,d,t}^i = D_\phi^{(k)}(\tau_t^i, \mathbf{0}, i)\big|_d.$$

The SAI $\chi_{l,d,t}^{i,(k)}$ for $l \in \{0, 1\}$ is then defined by

$$\chi_{l,d,t}^{i,(k)} = \mathbb{I}\left[\left\|\hat{s}_{l,d,t}^i - s_{d,t}\right\|^2 \leq \alpha\right], \quad (2)$$

where $\mathbb{I}$ is the indicator function and $\alpha$ is a reconstruction threshold. We aggregate $\chi_{l,d,t}^{i,(k)}$ to obtain message-dependent

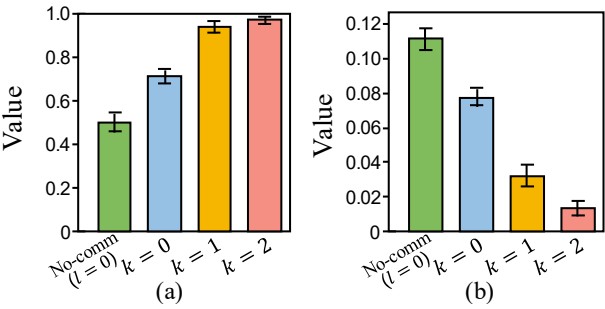

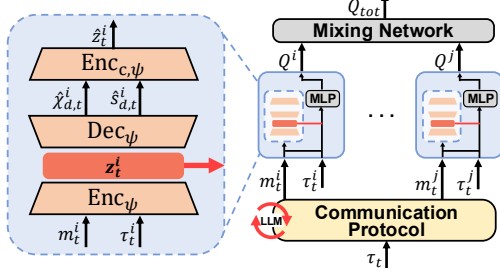

*Figure 5.* Overall framework of the proposed LMAC

*Figure 4.* Iteration-wise evaluation of SAI: (a) *recovery success rate* and (b) *knowledge imbalance*.

recovery statistics and convert them into an iteration-specific feedback instruction $\tilde{x}^{(k+1)}$ that guides the next protocol update toward the step-wise objective. As shown in Fig. 3(c), each feedback instruction contains a data-analysis instruction and a feedback-generation instruction, prompting the LLM to analyze how well the current messages satisfy the objective and to propose protocol refinements accordingly. Finally, given feedback instruction $\tilde{x}^{(k+1)}$, the LLM generates the feedback sentence $c^{(k+1)}$.

As shown in Fig. 1, the analysis criterion depends on the iteration. In the recovery enhancement step, we summarize agent-wise recovery for each state dimension using the *recovery success rate* $\mathbb{E}_t\left[\chi_{l,d,t}^{i,(0)}\right]$, which quantifies how well agent $i$ reconstructs dimension $d$ under $f_C^{(0)}$. In the imbalance mitigation step, we summarize *inter-agent knowledge imbalance* using $\mathrm{Var}_i\left[\chi_{l,d,t}^{i,(1)}\right]$, which quantifies how differently agents reconstruct dimension $d$ under $f_C^{(1)}$. The feedback-generation instruction uses these summaries to evaluate the current messages and propose refinements aligned with the step-wise objective.

**Output Communication Protocol.** Overall, with the proposed input prompt and feedback instructions, the communication protocol $f_C^{(k)}$ is iteratively improved in the code format shown in Fig. 3(b) through the following process.

- **Protocol initialization** ($k = 0$)**:** Using the task prompt $x$, we generate an initial executable protocol $f_C^{(0)}$ that maps agents' observations to minimal yet semantically meaningful messages.
- **Recovery enhancement** ($k = 1$)**:** To improve reconstruction accuracy, we use the recovery success rate to generate feedback $c^{(1)}$ via $\tilde{x}^{(1)}$, refining the protocol into $f_C^{(1)}$ by adding task-relevant information where needed.
- **Imbalance mitigation** ($k = 2$)**:** To reduce discrepancies in state awareness, we use the knowledge imbalance to generate feedback $c^{(2)}$ via $\tilde{x}^{(2)}$, refining the protocol into $f_C^{(2)}$ so that recovery becomes more consistent.

Through this process, we obtain the final communication protocol sequence $f_C = (f_C^{(0)}, f_C^{(1)}, f_C^{(2)})$, which progres-

sively adds messages to improve both recovery accuracy and uniformity. We limit refinement to two steps, since experiments with additional iterations ($k > 2$) in Appendix D.1 show negligible further gains. To assess the refinement behavior, Fig. 4 reports iteration-wise averages of the recovery success rate $\mathbb{E}_t\left[\chi_{1,d,t}^{i,(0)}\right]$ and the knowledge imbalance $\mathrm{Var}_i\left[\chi_{1,d,t}^{i,(1)}\right]$. The recovery success rate increases steadily, while knowledge imbalance decreases, indicating improved information recovery and more uniform state awareness across agents. Additional details on offline dataset construction, full prompt descriptions, and decoder training are provided in Appendix A.

### 4.3. Meta-Cognitive Representation Learning for MARL Framework

With the final communication protocol $f_C$ that improves state awareness, we apply this pre-designed protocol during MARL training to generate agent-specific messages $m_t^i = \left(m_t^{i,(0)}, m_t^{i,(1)}, m_t^{i,(2)}\right) = f_C(\tau_t)$. We then propose LMAC, a MARL framework that integrates the LLM-designed protocol into CTDE training. We adopt QMIX (Rashid et al., 2020) as the base learner, while the framework readily extends to other CTDE paradigms such as VDN (Sunehag et al., 2017) and QPLEX (Wang et al., 2020). Rather than feeding raw messages directly to agents, we introduce an encoder–decoder pair $(\mathrm{Enc}_\psi, \mathrm{Dec}_\psi)$ parameterized by $\psi$ and define the latent representation as $z_t^i = \mathrm{Enc}_\psi(\tau_t^i, m_t^i)$, which is used for state inference and decision-making.

We reuse the SAI in Eq. 2 as a dynamic supervision signal during online CTDE learning. Unlike the offline protocol-design stage, $\chi_{d,t}^i$ is computed per training batch using the training-time global state available to the centralized learner. We jointly train the encoder–decoder to reconstruct $s_t$ and predict $\chi_{d,t}^i$, encouraging a meta-cognitive representation that distinguishes reliable knowledge from uncertainty. To further suppress irrelevant information, we add a cycle-consistency loss inspired by Zhu et al. (2017), decoding and re-encoding the latent via an auxiliary encoder $\mathrm{Enc}_{c,\psi}$ as $\hat{z}_t^i = \mathrm{Enc}_{c,\psi}\left(\mathrm{Dec}_\psi(z_t^i)\right)$ and training it to reconstruct $z_t^i$. Because information that cannot be preserved through decoding and re-encoding is penalized, this constraint discourages redundant content and encourages $z_t^i$ to retain

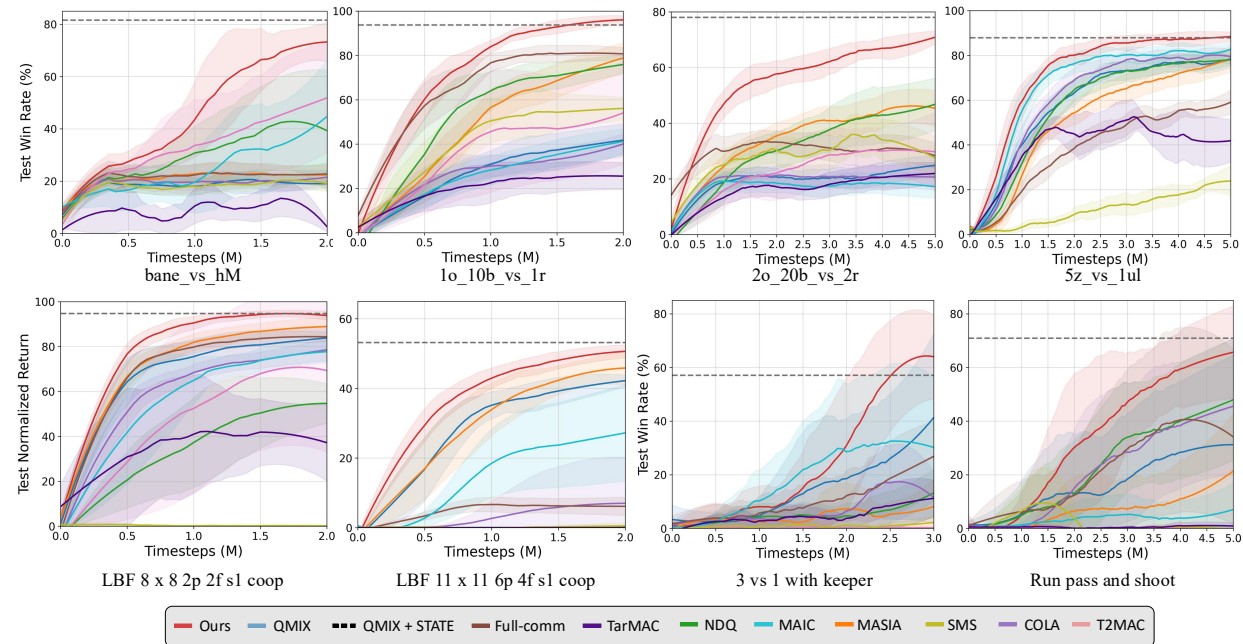

*Figure 6.* Performance comparison in various MARL benchmarks

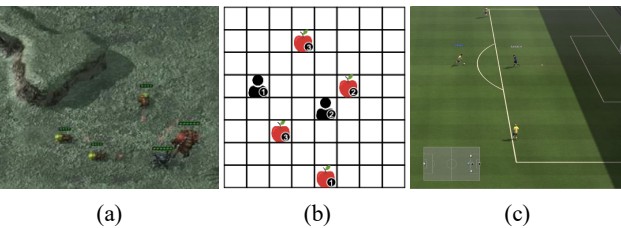

*Figure 7.* MARL Benchmarks used in our experiments: (a) SMAC-Comm, (b) LBF, and (c) GRF

only reconstructable, task-relevant features aligned with reconstruction reliability. Finally, we incorporate $z_t^i$ into the individual utilities as $Q^i(\tau_t^i, z_t^i)$ and optimize the joint action-value $Q^{\text{tot}}$ via TD learning. In summary, the overall LMAC framework is shown in Fig. 5, with additional details provided in Appendix A.

# 5. Experiments

In this section, we evaluate the proposed method on four benchmark environments shown in Fig. 7: Star-Craft Multi-Agent Challenge with Communication (SMAC-Comm) (Samvelyan et al., 2019), a communication-intensive variant of StarCraft II evaluated on `bane_vs_hM`, `1o_10b_vs_1r`, `5z_vs_1ul`, and `2o_20b_vs_2r`, the last of which is designed to evaluate scalability with a larger number of agents; Level-Based Foraging (LBF) (Papoudakis et al., 2020), a cooperative foraging task with settings `8x8-2p-2f-s1-coop` and `11x11-6p-4f-s1-coop` ($n \times n$: grid size, $p$: agents, $f$: fruits, $s$: sight range), Google Research Football (GRF) (Kurach et al., 2020), a cooperative soccer game with scenarios `3_vs_1_with_keeper` and `run_pass_and_shoot`, and SMACv (Ellis et al.,

2023), a more stochastic StarCraft II benchmark that randomizes unit types, positions, and team compositions across episodes. We first compare performance against other communication baselines, then analyze how step-wise protocols contribute to cooperation. Unless otherwise specified, experiments use `gpt-4.1-2025-04-14` as the backbone LLM, with variants included in ablation studies. The task description $I_T$ used in the prompts is based on the original environment documentation provided by the benchmark designers (Samvelyan et al., 2019; Wang et al., 2019; Papoudakis et al., 2020; Kurach et al., 2020), with only slight editing for clarity and consistency. For each task, we construct the offline transition dataset $\mathcal{B}$ by collecting 5k trajectories during QMIX training. All results are averaged over 5 random seeds with standard deviations, and further details are provided in Appendix B.

## 5.1. Performance Comparison

To validate our approach, we compare LMAC against a comprehensive set of communication-based MARL methods. In addition to the broadcasting and agent-wise approaches analyzed in Sec. 4.1 (i.e., **FullComm**, **MASIA** (Guan et al., 2022), **TarMAC** (Das et al., 2019), **SMS** (Xue et al., 2022), and **T2MAC** (Sun et al., 2024)), we include our backbone algorithm **QMIX** (Rashid et al., 2020) and **QMIX+State**, which provides global state information during execution as an upper-bound reference. We further evaluate **NDQ** (Wang et al., 2019), which reduces communication cost via decomposable value functions; **MAIC** (Yuan et al., 2022), which generates incentive messages to shape teammates' utilities; and **COLA** (Xu et al., 2023), which improves coordination via inter-agent consensus. All baselines are evaluated using

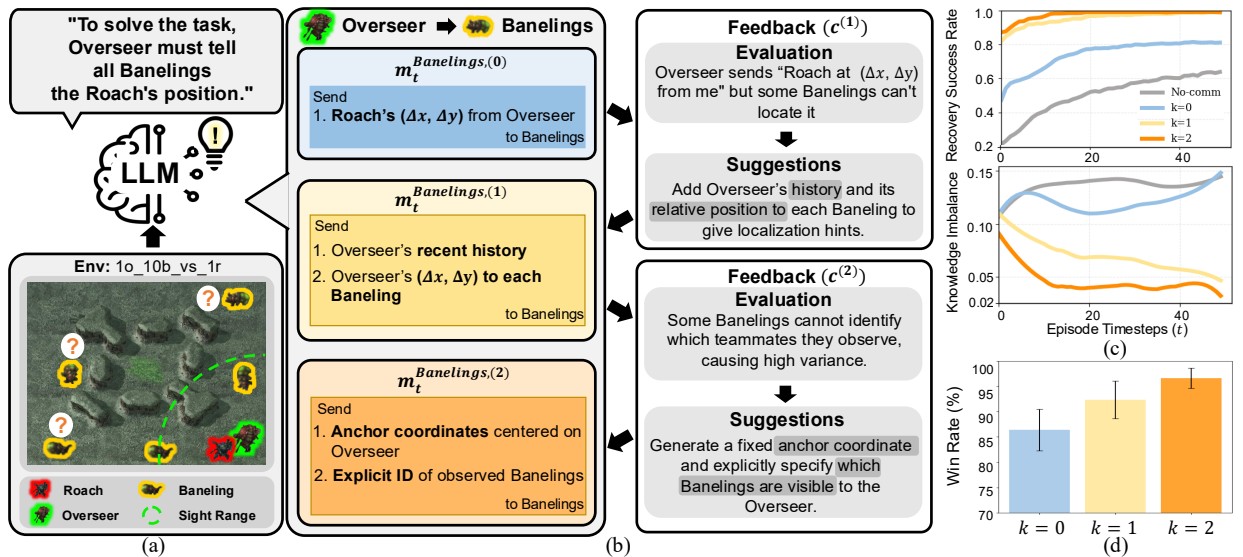

*Figure 8.* Protocol refinement analysis on SMAC `1o_10b_vs_1r`: (a) Task scenario with an Overseer, a Roach, and Banelings under partial observability, (b) protocol messages and corresponding feedback at each step $k$, (c) trajectory-level *recovery success rate* and *inter-agent knowledge imbalance* with ($k = 0, 1, 2$) or without messages (No-comm), and (d) average win rates across steps.

author-released implementations. For our method, we report results with the best-performing threshold $\alpha$, and full hyperparameter settings and additional experimental details are provided in Appendix B.

Fig. 6 reports success rates across the three benchmark environments. On SMAC-Comm, our method converges faster and achieves higher final success rates across all four scenarios, with particularly large gains on `bane_vs_hM` and the large-scale `2o_20b_vs_2r`. This suggests strong scalability when state recovery is difficult or the number of agents increases. Notably, our results nearly match the upper-bound QMIX+State, indicating that the designed protocols capture critical state information. On LBF, we observe similar trends: our method learns faster and consistently reaches higher final performance, again approaching QMIX+State, supporting that refined communication enables sufficient state reconstruction for coordinated behavior.

On GRF, our method not only surpasses all baselines but also outperforms QMIX+State in final success rates. Because GRF features high-dimensional observations, directly providing the full state can increase input dimensionality and slow learning. In contrast, our latent learning compresses messages into compact, task-relevant features, enabling faster convergence and stronger cooperative strategies. Overall, these results show that our framework consistently improves both learning speed and final performance across diverse communication-required MARL settings, enabling agents to exploit information more effectively.

To evaluate the adaptability of our method to a more challenging setting characterized by high stochasticity, we conducted additional experiments on SMACv2 (Ellis et al., 2023). By randomizing unit configurations in every episode,

*Table 1.* Test win rates (%) at 3M environment steps in SMACv2: `terran_5_vs_5`, `protoss_5_vs_5`, and `zerg_5_vs_5`.

| Algorithm | Scenario | | |
| --- | --- | --- | --- |
| | **terran** | **protoss** | **zerg** |
| QMIX | $61.69 \pm 5.4$ | $48.44 \pm 4.3$ | $34.79 \pm 4.7$ |
| FullComm | $18.44 \pm 1.7$ | $16.93 \pm 3.1$ | $6.83 \pm 1.8$ |
| TarMAC | $29.68 \pm 2.2$ | $22.56 \pm 2.1$ | $15.47 \pm 2.2$ |
| NDQ | $59.20 \pm 5.2$ | $48.03 \pm 4.2$ | $38.75 \pm 2.9$ |
| MAIC | $63.80 \pm 3.4$ | $51.93 \pm 2.4$ | $38.59 \pm 2.8$ |
| MASIA | $54.72 \pm 7.0$ | $32.43 \pm 3.7$ | $34.22 \pm 2.6$ |
| SMS | $34.00 \pm 3.2$ | $27.00 \pm 5.8$ | $13.53 \pm 3.6$ |
| COLA | $10.29 \pm 9.5$ | $0.01 \pm 0.0$ | $0.19 \pm 0.3$ |
| T2MAC | $61.67 \pm 2.5$ | $48.16 \pm 5.6$ | $35.09 \pm 3.4$ |
| QMIX+STATE | $64.77 \pm 2.79$ | $56.40 \pm 2.33$ | $40.06 \pm 3.39$ |
| **LMAC (Ours)** | $\mathbf{67.87 \pm 2.77}$ | $\mathbf{57.96 \pm 4.02}$ | $\mathbf{42.18 \pm 4.37}$ |

SMACv2 introduces severe distribution shifts that prevent fixed-scenario memorization. In this rigorous environment, as shown in Table 1, LMAC consistently outperforms existing communication baselines across all three scenarios. Notably, LMAC even surpasses QMIX+State, suggesting that the LLM-designed protocol facilitates efficient exchange of critical information and filters task-relevant features more effectively than directly utilizing the raw global state under high stochasticity.

### 5.2. Trajectory Analysis

To analyze how our framework produces communication protocols that improve information recovery and reduce imbalance, we perform a trajectory-level analysis on the SMAC `1o_10b_vs_1r` map, summarized in Fig. 8. In (a), the task requires 10 Banelings (10b) to quickly converge on

*Table 2.* Ablation studies on SMAC-Comm: (a) component evaluation, (b) LLM variants, and (c) reconstruction threshold $\alpha$.

| Setting | Avg. Win Rate (%) | LLM | Avg. Win Rate (%) | $\alpha$ | Avg. Win Rate (%) |
|---|---|---|---|---|---|
| w/o Cons | $66.5 \pm 2.1$ | GPT | $\mathbf{82.9 \pm 1.9}$ | 0.0005 | $77.2 \pm 2.1$ |
| w/o SAI | $76.6 \pm 5.6$ | GPT-mini | $79.8 \pm 1.5$ | 0.002 | $79.3 \pm 2.8$ |
| $k = 0$ | $68.5 \pm 3.8$ | GPT-o1-mini | $81.8 \pm 2.9$ | 0.005 | $80.5 \pm 1.7$ |
| $k = 1$ | $77.8 \pm 2.2$ | Claude | $81.9 \pm 2.6$ | 0.05 | $\mathbf{82.9 \pm 1.9}$ |
| $k = 2$ (Ours) | $\mathbf{82.9 \pm 1.9}$ | Gemini | $80.8 \pm 1.6$ | 0.5 | $80.2 \pm 3.2$ |

| (a) Component evaluation | (b) LLM variants | (c) $\alpha$ comparison |
|---|---|---|

a Roach (1r), with the Overseer (1o) providing positional cues. Because absolute positions are not available in raw observations, the LLM infers that agents must recover both the Roach's location and each Baneling's absolute position; otherwise, coordination delays reduce damage. In (b), we visualize protocol evolution.

At $k = 0$, the Overseer broadcasts the Roach's relative offset $(\Delta x, \Delta y)$ from itself, which enables partial localization but remains insufficient without identifying the Overseer's position. At $k = 1$, feedback indicates that the Overseer's position is difficult to infer, and the protocol is refined to include the Overseer's relative position and recent history as localization cues. At $k = 2$, variance-based feedback shows that some Banelings still cannot disambiguate which teammates they observe, leading the protocol to introduce a fixed anchor coordinate centered on the Overseer and explicit IDs for observed agents. The final protocol therefore shares the Roach's relative position, the anchor, and teammate IDs, enabling consistent absolute recovery across agents. In (c), the *recovery success rate* increases in step 1 ($k = 1$), while the *inter-agent knowledge imbalance* decreases in step 2 ($k = 2$). In (d), learning performance improves across steps as agents achieve shared localization and coordinate attacks more reliably. Overall, these results demonstrate that LMAC produces step-wise refined, task-relevant messages whose refinement directly improves recovery and reduces imbalance, consistent with the intended protocol design. Similar patterns appear in other environments, with additional trajectory analyses provided in Appendix C.

### 5.3. Ablation Study

We conduct three ablation studies on SMAC-Comm to validate key components of our framework: (a) component evaluation, (b) comparison across LLM variants, and (c) sensitivity to the reconstruction threshold $\alpha$. We further analyze whether LMAC depends on detailed prompt descriptions and whether it generalizes across different value decomposition backbones.

**Component Evaluation.** To analyze the effect of each component, we compare the performance of protocols obtained at different refinement stages ($k = 0, 1, 2$), as well as two variants of our framework: one without the consis-

tency loss ('w/o Cons') and one without SAI reconstruction ('w/o SAI'). As shown in Table 2 (a), performance steadily improves as the refinement step progresses, consistent with our earlier findings that state recovery and balance improve with each stage. Removing the consistency loss causes a clear drop in performance, demonstrating that eliminating redundant information in messages is crucial for learning. Similarly, removing SAI also degrades performance, confirming that knowing not only the recovered state but also its reliability is essential for effective cooperation. Overall, these results show that every component of our framework contributes substantially to performance.

**LLM Variants.** Table 2 (b) evaluates our method with different LLMs, including GPT-4.1, GPT-4.1-mini, o1-mini, Gemini-2.5-Flash, and Claude-Opus. Results show that all recent LLMs achieve strong performance, with GPT providing the highest success rates. However, smaller and more efficient models (e.g., GPT-4.1-mini, o1-mini) still perform competitively, demonstrating that the key driver of performance is the multi-step refinement process rather than reliance on a particular model. This refinement procedure further encourages consistency in the resulting communication protocols, ensuring that our framework remains broadly applicable across different LLMs.

**Reconstruction Threshold $\alpha$.** Table 2 (c) analyzes the effect of the reconstruction threshold $\alpha$ used in constructing SAI. An overly small $\alpha$ makes the criterion overly strict, causing most dimensions to be judged unrecoverable and leading to excessive message generation. While performance is maximized at $\alpha = 0.05$, results remain stable for larger values, suggesting that our method is not highly sensitive to this parameter. This robustness indicates that our framework can be applied without heavy hyperparameter tuning.

**Prompt Simplification.** Since the LLM prompt may assume structured environment descriptions and protocol design requirements, we examine whether LMAC remains effective when the overall prompt information is provided in a more compact form. For this analysis, we evaluate a simplified variant on the SMAC-Comm scenarios `1o_10b_vs_1r` and `bane_vs_hM`, where effective communication strongly depends on positional and unit-status information. In this

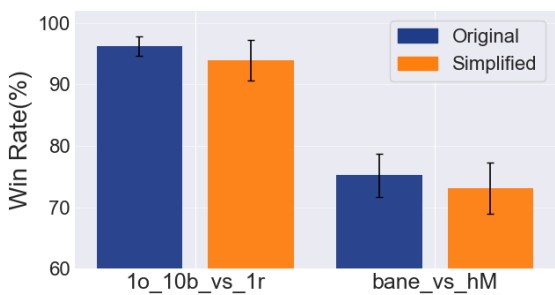

*Figure 9.* Win rate (%) under simplified prompt.

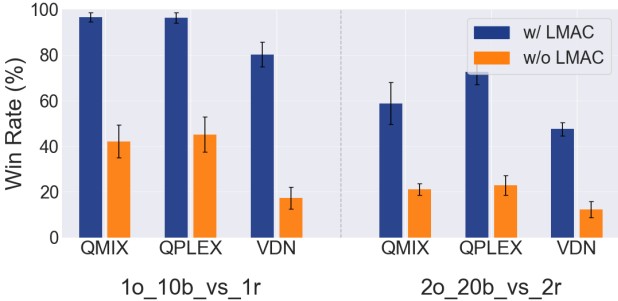

*Figure 10.* Generality analysis across CTDE backbones on SMAC-Comm `1o_10b_vs_1r` and `2o_20b_vs_2r`.

variant, the task description $I_T$ is reduced to a coarse summary of allied and enemy unit status and spatial configuration, while the observation description is given only at the chunk level rather than dimension-wise. The protocol design instruction $I_P$ is also simplified to retain only the core communication objective and required output format. Examples are provided in Appendix A.3. As shown in Fig. 9, this simplified variant yields only minor performance differences in both scenarios, suggesting that LMAC does not rely heavily on detailed prompt structuring or environment-provided semantic descriptions.

**Generality of LMAC.** We further evaluate the generality of LMAC by combining it with different value decomposition backbones, including VDN, QMIX, and QPLEX. As shown in Fig. 10, LMAC provides consistent performance gains across these backbones, indicating that it is not specific to QMIX. This trend is also observed in the larger `2o_20b_vs_2r` scenario, suggesting that the proposed communication framework remains applicable as the number of agents increases. These results support the generality of LMAC across both CTDE backbones and environment scales.

**Further Analysis.** We provide additional analyses to validate LMAC. Appendix D.1 analyzes the effect of the refinement steps and shows that performance saturates beyond $k = 2$, motivating our default choice of $k = 2$. Appendix D.2 shows that LLM-designed messages outperform baseline message designs under the same LMAC training pipeline. Appendix D.3 demonstrates robustness to both the initial offline data source and moderate environment changes. Finally, Appendix D.4 and Appendix D.5 show that LMAC remains effective under reduced communication capacity, while incurring only modest computational overhead and limited offline LLM usage.

## 6. Limitations and Future Work

Our framework performs well as intended, but it has a few limitations. First, protocol refinement introduces additional overhead from auxiliary decoder training and offline LLM calls, although our complexity analysis in Appendix D.5 shows that this cost remains modest in practice. Second,

the protocol quality can depend on the reasoning capability of the chosen LLM, although our ablations show robust performance across multiple LLM variants.

A further direction is to extend LMAC beyond structured descriptions. Image observations would require mapping raw visual inputs into semantic attributes, such as object identities and spatial relations, as studied in visual representation learning (Wen et al., 2022; Yoon et al., 2023; Didolkar et al., 2025). Based on such representations, LMAC could be extended toward semantic-attribute communication for unstructured visual observations.

## 7. Conclusion

We propose LMAC, an LLM-driven communication framework for cooperative MARL that promotes unified state awareness under partial observability. LMAC iteratively refines an executable, agent-wise protocol via multi-step feedback, explicitly improving task-relevant state recovery while reducing inter-agent knowledge imbalance. The resulting protocol is integrated into CTDE learning through a meta-cognitive latent module that supports state reconstruction and reliability calibration, with a cycle-consistency constraint encouraging compact and reconstructable representations. Across multiple benchmarks, LMAC consistently improves both coordination and team performance over prior communication methods.

## Acknowledgment

This work was supported partly by the Institute of Information & Communications Technology Planning & Evaluation (IITP) grant funded by the Korea government (MSIT) (No. RS-2022-II220469, Development of Core Technologies for Task-oriented Reinforcement Learning for Commercialization of Autonomous Drones), (No. RS-2025-25442824, AI Star-Fellowship Program (UNIST)), and (No. RS-2020-II201336, Artificial Intelligence Graduate School Support (UNIST)), and partly by the National Research Foundation of Korea (NRF) grant funded by the Korea government (MSIT) (No. RS-2025-23523191, LLM-Based Multi-Agent Reinforcement Learning for End-to-End Large Autonomous Swarm Control).

## Impact Statement

This paper presents work whose goal is to advance the field of Machine Learning. There are many potential societal consequences of our work, none of which we feel must be specifically highlighted here.

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

# A. Implementation Details

This section provides the implementation details of our refinement process. Appendix A.1 describes the construction of prompts used in different refinement steps. Appendix A.2 presents examples of LLM outputs for both protocol generation and feedback. Finally, Appendix A.4 details the training objectives of LMAC, including the reconstruction, meta, and consistency losses. For each prompt, we highlight task-specific and step-specific content in blue, provide corresponding examples in the figures, and include full details in the code implementation.

## A.1. Prompt Construction for Refinement Process

**Input prompt** $x$. The input prompt $x$, designed to guide the generation and refinement of communication protocols, is formulated as a tuple $x = (\mathcal{I}_T, \mathcal{I}_P)$. This composite structure ensures that LLM receives both the semantic understanding of the specific environment and the rigorous constraints required for protocol implementation. Before generating the protocol, we implicitly guide the LLM through a CoT (Wei et al., 2022) process to identify task-critical state dimensions, filtering out irrelevant noise to focus on essential information for state reconstruction. Task description $\mathcal{I}_T$, provides the environment-specific context. It is structured to inform the agent about its goal, the global state space, and the local observation space. The template for $\mathcal{I}_T$ is presented below:

---

**Task Description ($\mathcal{I}_T$)**

**Task Description and Environment Characteristics:**
1. Task Description:
{task_description}
2. State Information:
{state_description}
3.Observation Information:
{observation_description}

---

Within this template, the placeholders are dynamically populated based on the specific scenario. The {task_description} outlines the overall objective, derived from original scenario definitions in benchmarks such as SMAC-Comm (Samvelyan et al., 2019; Wang et al., 2019), SMACv2 (Ellis et al., 2023), LBF (Yuan et al., 2022), and GRF (Kurach et al., 2020). The {state_description} and {observation_description} provide a semantic breakdown of the global state and local observation vectors, respectively, mapping numerical dimensions into natural language following the methodology of Wang et al. (2024). To illustrate how these abstract descriptions are materialized in practice, we present representative examples from the SMAC-Comm `1o_10b_vs_1r` scenario in Fig. A.1.

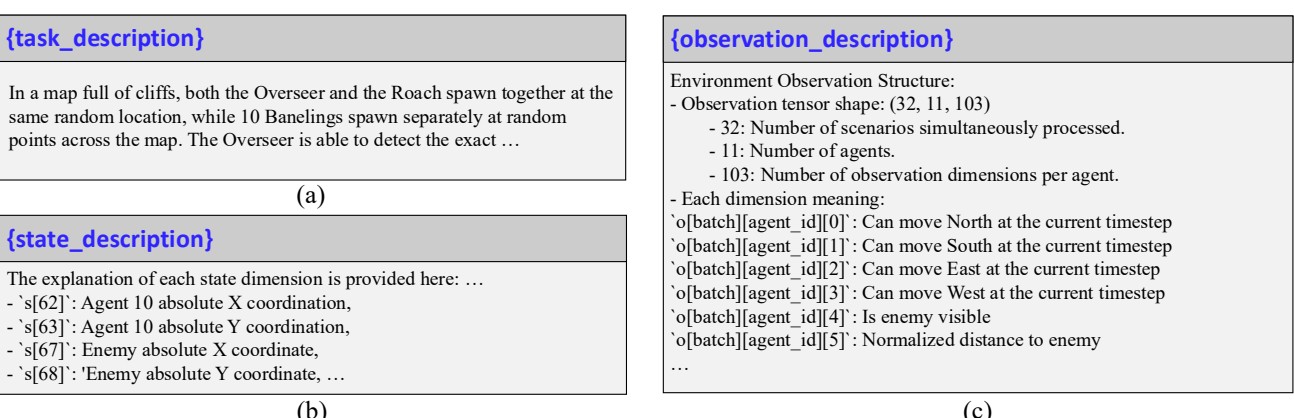

*Figure A.1.* Illustrative examples of (a) task description, (b) state description, and (c) observation description in the SMAC-Comm `1o_10b_vs_1r` scenario.

Protocol design instruction $\mathcal{I}_P$ provides the environment-agnostic protocol design instruction, specifying the core communication principles and constraints for generating executable Python code:

**Protocol Design Instruction ($\mathcal{I}_P$)**

**Communication Design Key Principles**:

1. State Reconstruction & Knowledge Gap Bridging:
- Analyze Semantic Relationship: Before designing, map local observations to global state variables to identify what is missing (Knowledge Gap).
- Target Partial Observability: In POMDPs, global states are hidden. Do not access them directly. Instead, share correlated local features that allow others to infer the missing context.

2. Uniqueness, Sufficiency & Compactness:
- Share only essential information not already known or easily inferred by others.
- Ensure sufficiency for coordination while strictly minimizing redundancy.

3. Contextual and Interaction-Aware:
- Prioritize self-perceived behavioral data (e.g., movement possibilities, recent actions) to compensate for partial visibility.

4. Explicitness and Clarity:
- Avoid abstraction; critical task information must be explicit and interpretable.

5. Structured Output:
- Output shape: ({batch, agents, obs_dim} + message_dim).

6. Communication Protocol:
- Messages must be distributed and concatenated into other agents' observations.
- Do NOT include the message in the sender's own observation.

7. Computational Efficiency:
- No trainable components; minimize loops for batch efficiency.

**Task**: Design a protocol using the identified critical dimensions to improve coordination and state recovery.
**Required Python Functions**:

1. `message_design_instruction()`:
- Returns a string explaining how the message aids global state reconstruction using critical dimensions.

2. `communication(o)`:
- Input: Observation `o`.
- Output: Enhanced observation with messages that reduce state uncertainty for other agents.
- Logic: Extract key information from `o` and format it for others' consumption.

**Constraints**:
- Executable, integration-ready Python code.
- Vectorized operations only (minimize for-loops) for efficiency.

Let's think step by step. Below is an illustrative example of the expected output:
```python
import torch as th
def message_design_instruction():
# Explain how this protocol aids state reconstruction via critical dimensions
return message_description
def communication(o):
# Implement protocol logic using vectorized operations.
# Input o: {obs_shape}, Output: {obs_shape} + message_dim
# Ensure device consistency.
return messages_o
```

**Feedback Instructions** $\tilde{x}^{(1)}$, $\tilde{x}^{(2)}$. As described in Section 4.2, the feedback instructions serve as the bridge between quantitative SAI metrics and qualitative code generation. Unlike the fixed feedback mechanism in standard Reflexion, our protocol employs distinct instructions for each optimization step: $\tilde{x}^{(1)}$ for Recovery enhancement and $\tilde{x}^{(2)}$ for Imbalance mitigation. The feedback instruction follows a structured template comprising a step description, the previous communication protocol code, and criterion-based performance data. This inclusion of the previous protocol code $f_C^{(k)}$ allows the LLM to directly map the quantitative weaknesses found in the criterion data to specific lines of code that require modification. The placeholders within brackets are dynamically populated to align with the specific objectives of Step 1 ($k = 1$) and Step 2 ($k = 2$).

---

### Step-wise Feedback Instruction

You are an analysis agent tasked with improving communication strategies in a multi-agent reinforcement learning (MARL) system.

**Data-Analysis Instruction**

1. Step Information & Guidelines:
Conduct your analysis based on the objectives below.
{step_description}

2. Previous Protocol:
{previous_comm_protocol $f_C^{(k)}$}

3. Criterion Data Analysis:
Analyze the auxiliary decoder results below based on the 'Guidelines' provided.
{criterion}

**Feedback Generation Instruction**
Based on your analysis, generate the structured feedback.

**Expected Output Format (JSON):**
```
{
"Evaluation":  "Synthesize your analysis results.  Explicitly mention the gaps
identified using the Step analysis method",
"Missing_Information_Hypothesis":  "Hypothesis about what specific information
is missing or inadequately communicated.",
"Improvement_Suggestions":  "Specific, actionable suggestions to modify the
communication content/structure.  These must directly address the identified
gaps to achieve the step goal"
}
```

---

The dynamic components, {step_description} and {criterion}, adapt the LLM's reasoning to each optimization step. The {step_description} injects the step-specific objective: Step 1 focuses on recovery enhancement to maximize individual recovery rates by identifying agent-level ignorance, while Step 2 targets imbalance mitigation to minimize inter-agent knowledge gaps by detecting information asymmetry. The specific instructions for each step are detailed below.

---

### {step_description}

**Recovery enhancement (Step 1)**

- **Step**: 1 (Recovery Enhancement)
- **Goal**: Ensure each agent can recover state dimensions accurately.
- **Objective**: Individual agent enhancement through improved communication.
- **Focus Areas**:
  - Improve individual agent prediction accuracy for critical state dimensions.
  - Ensure each agent receives information needed for better state inference.
  - Address agent-specific limitations in state dimension recognition.

---

**Imbalance mitigation (Step 2)**

- **Step**: 2 (Imbalance Mitigation)
- **Goal**: Achieve consistent state recovery across all agents.
- **Objective**: Address inconsistencies and reduce information asymmetry.
- **Focus Areas**:
  - If one agent can recover a state dimension, all agents should be able to do so consistently.
  - Address information imbalance (high variance) across agents.
  - Focus on dimensions where agents show inconsistent prediction performance.

---

The {criterion} field provides the quantitative basis for the step-wise analysis by presenting the aggregated SAI results in a structured JSON format. In Step 1, it reports the *recovery success rate* $\mathbb{E}_t[\chi_{l,d,t}^{i,(0)}]$, as illustrated in Fig. A.2(a), to facilitate the identification of local reconstruction failures. In Step 2, it reports the *inter-agent knowledge imbalance* $\mathrm{Var}_i[\chi_{l,d,t}^{i,(1)}]$, as shown in Fig. A.2(b), enabling the diagnosis of inter-agent inconsistencies.

**Recovery Success Rate** $E_t\left[\chi_{l,d,t}^{i,(0)}\right]$

```
"dimension": "s[62] - Agent 10 absolute X coordination",
  "with_communication": {
    "agent_success_rates": [0.64, 0.56, … , 0.65, 0.95]
  },
  "without_communication": {
    "agent_success_rates": [0.52, 0.47, … , 0.39, 0.94]
  }

…
"dimension": "s[63] Agent 10 absolute Y coordination",
  "with_communication": {
    "agent_success_rates": [0.98, 0.98, … , 0.97, 1.0]
  },
  "without_communication": {
    "agent_success_rates": [0.38, 0.37, … , 0.39, 1.0]
  }
…
```

(a)

**Inter-agent Knowledge Imbalance** $Var_i\left[\chi_{l,d,t}^{i,(1)}\right]$

```
"dimensions":"s[62] - Agent 10 absolute X coordination",
  "with_communication": {
          "early": [0.407, … , 0.775], variance: [0.033],
      …
          "mid": [0.791, … , 0.978], variance: [0.013]
      …
    "late": [0.766, … ,1.0], variance: [0.015]
  },

  "without_communication": {
    "early": [0.219, … , 0.769], variance: [0.363],
      …
        "mid": [0.575, … , 0.981], variance: [0.243],
          …
    "late": [0.716, … , 1.0], variance: [0.143]
```

(b)

*Figure A.2.* Illustrative examples of (a) *recovery success rate* $\mathbb{E}_t[\chi_{l,d,t}^{i,(0)}]$ and (b) *inter-agent knowledge imbalance* $\mathrm{Var}_i[\chi_{l,d,t}^{i,(1)}]$.

## A.2. Detailed Example of Communication Protocol

Since protocol generation relies on high-dimensional tensors, the LLM may not align dimensions correctly in one attempt. We therefore adopt a Reflexion-based mechanism, where error messages are fed back for iterative correction. More details of iterative correction are provided in Appendix D.5 Expressing the protocol in code allows direct verification of transmitted information, while the accompanying message design instruction makes the communication process interpretable even to non-experts. Here is the example of the communication protocol in SMAC-Comm `1o_10b_vs_1r` scenario. Specifically, this represents the initial generated protocol $f_C^{(0)}$. It is worth noting that subsequent iterations, such as $f_C^{(1)}$ and $f_C^{(2)}$, are generated using the identical format.

**Example: Communication Protocol $f_C^{(0)}$**

```python
import torch as th

def communication(o):
    """
    o: (batch, 11, 103)
    Returns: (batch, 11, 117)
    """
    # Device safety
    device = o.device
    batch_size = o.shape[0]
    n_agents = o.shape[1]
    obs_dim = o.shape[2]
    message_dim = 14

    # 1. Get Overseer's observation (agent index 10)
    overseer_obs = o[:, 10, :]  # (batch, 103)

    # 2. Extract required info from Overseer
    # These are all (batch,)
    relative_x = overseer_obs[:, 6].unsqueeze(1)      # (batch, 1)
    relative_y = overseer_obs[:, 7].unsqueeze(1)      # (batch, 1)
    is_visible = overseer_obs[:, 4].unsqueeze(1)      # (batch, 1)

    # 3. Create sender_id one-hot (always index 10)
    sender_id = th.zeros(batch_size, 11, device=device)
    sender_id[:, 10] = 1.0  # (batch, 11)

    # 4. Assemble message: (batch, message_dim)
    overseer_message = th.cat([relative_x, relative_y, is_visible, sender_id], dim=1)
        # (batch, 14)

    # 5. Broadcast message to all agents
    # For agents 0-9: receive Overseer's message. For agent 10: receive zeros.
    messages = th.zeros(batch_size, n_agents, message_dim, device=device)

    # For Banelings (indices 0~9), set their message to Overseer's message
    messages[:, 0:10, :] = overseer_message.unsqueeze(1).expand(-1, 10, -1)
    # For Overseer (index 10), message remains zeros

    # 6. Concatenate messages to observations
    messages_o = th.cat([o, messages], dim=2)  # (batch, 11, 117)
    return messages_o
```

## A.3. Example of Simplified Prompt

Here, we provide example simplified prompts used in the prompt simplification analysis in Sec. 5.3. In the simplified task description $I_T$, the state information is described in a highly compact form that summarizes only the main attributes of the global state, while the observation information retains chunk-level structural guidance rather than a fully dimension-wise description. This is because effective communication protocol design still requires minimal information about where different types of observations are located. In the simplified protocol design instruction $I_P$, we retain only the core communication objective, namely information sharing among agents, and the required output format. Accordingly, the simplified prompt reduces detailed semantic annotations and design instructions while preserving the basic structural cues and output requirements needed for protocol generation.

---

**Simplified {State Description}**

**State Information**:

- The global state concatenates per-unit attributes absolute position, weapon cooldown, health, shield, unit type for allied units and enemy unit. All values are normalized to $[0, 1]$.

---

**Simplified {Observation Description}**

**Observation Information**:
- Observation tensor shape: (32, 11, 103)
- Each agent receives a flat 103-dim local observation `o[batch][agent_id]`.
  - dims `[0--3]`: movement availability
  - dims `[4--11]`: enemy observation (visibility, distance, position, health, shield, type)
  - dims `[12--81]`: ally observations (visibility, distance, position, health, shield, type per ally)
  - dims `[82--84]`: self (own) features (health, shield, type)
  - dims `[85--91]`: last action one-hot (7 dims)
  - dims `[92--102]`: agent ID one-hot (11 dims)

---

**Simplified Protocol Design Instruction ($\mathcal{I}_P$)**

**Communication Design Key Principles**:
- Design a communication protocol so that agents can reconstruct the global state from their local observations combined with received messages.

**Output Format**:
- Output shape: ({batch, agents, obs_dim} + message_dim).

**Communication Protocol**:
- Messages must be transmitted to other agents, ensuring they receive the information.
- The received message is then appended to the recipient's observation vector.
- Do NOT include the message in the sender's own observation.

**Task**: Design a protocol using the identified critical dimensions to improve coordination and state recovery.
**Required Python Functions**:

1. `message_design_instruction()`:
- Returns a string explaining how the message aids global state reconstruction using critical dimensions.

2. `communication(o)`:
- Input: Observation `o`.
- Output: Enhanced observation with messages that reduce state uncertainty for other agents.

- Logic: Extract key information from $\circ$ and format it for others' consumption.

**Constraints**:
- Executable, integration-ready Python code.
- Vectorized operations only (minimize for-loops) for efficiency.

Let's think step by step. Below is an illustrative example of the expected output:

```python
'''python
import torch as th
def message_design_instruction():
# Explain how this protocol aids state reconstruction via critical dimensions
return message_description
def communication(o):
# Implement protocol logic using vectorized operations.
# Input o: {obs_shape}, Output: {obs_shape} + message_dim
# Ensure device consistency.
return messages_o
'''
```

## A.4. Training Losses of LMAC

**Auxiliary Decoder Training.** We prepare a fixed trajectory dataset $\mathcal{B}$ to evaluate each step-specific protocol $f_C^{(k)}$. Specifically, $\mathcal{B}$ contains 5k trajectories per environment collected from the baseline QMIX (Rashid et al., 2020) training at 200k steps under $\epsilon$-greedy exploration across five seeds. For each step $k \in \{0, 1, 2\}$, the auxiliary decoder $D_\phi^{(k)}$ is trained to reconstruct the global state using an agent's trajectory and its message generated by $f_C^{(k)}$. Concretely, given agent $i$ at time $t$, we form $\hat{s}_{1,t}^i = D_\phi^{(k)}(\tau_t^i, m_t^{i,(k)}, i)$ and $\hat{s}_{0,t}^i = D_\phi^{(k)}(\tau_t^i, \mathbf{0}, i)$, which are used in Eq. 2 to compute $\chi_{l,d,t}^{i,(k)}$. The training objective for $D_\phi^{(k)}$ is the state reconstruction loss:

$$\mathcal{L}_{\text{aux}}(\phi) = \mathbb{E}_{(\tau, m^{(k)}, s) \sim \mathcal{B}}\Big[\|D_\phi^{(k)}(\tau_t^i, m_t^{i,(k)}, i) - s_t\|_2^2\Big]. \tag{A.1}$$

We implement $D_\phi^{(k)}$ as an autoencoder and optimize it with mini-batch SGD using the MSE loss in a supervised learning setup.

*Table A.1.* Hyperparameters used for training the auxiliary decoder $D_\phi^{(k)}$.

| Parameter | Value |
| --- | --- |
| Batch size | 32 |
| Dropout rate | 0.1 |
| Epochs | 1000 |
| Iterations per epoch | 10 |
| Hidden dimension | 64 |
| Latent dimension | 20 |
| Learning rate | 0.0005 |
| Optimizer | Adam |

**Representation and Policy Training.** The meta-cognitive representation learning module is implemented as an autoencoder architecture. The encoder $Enc_\psi$ incorporates an attention mechanism, where the query is formed from the current observation and received messages, while the key and value are derived from the trajectory $\tau_t^i$. This allows the latent representation $z_t^i$ to capture how the current observation and message attend to $\tau_t^i$. The decoder $Dec_\psi$ takes $z_t^i$ as input and produces two outputs: (i) a state estimate $\hat{s}_t^i$ and (ii) a prediction of the SAI $\hat{\chi}_{d,t}^i$, where the target label $\chi_{d,t}^i$ is computed online per training batch using the centralized global state (Eq. 2). We train the representation using the replay buffer $\mathcal{D}$ via (a) a reconstruction objective that jointly learns $\hat{s}_t^i$ and $\hat{\chi}_{d,t}^i$ and (b) a cycle-consistency regularizer that suppresses encoding of irrelevant factors in $z_t^i$ by requiring the decoded output to be re-encoded back to the same latent code. Formally, the two

---

**Algorithm 1** LLM-driven Multi Agent Communication (LMAC)

---

1: **Initialize:** task prompt $x = (\mathcal{I}_T, \mathcal{I}_P)$, reasoning tokens $z^{(0)}$, auxiliary decoder $\phi$, meta-cognitive representation learning module $\psi$, Q-network $\omega$ and its target network $\omega^-$
2: **for** $k = 0, 1, 2$ **do**
3:    Generate communication protocol $f_C^{(k)} \sim f_\theta^{\mathrm{LLM}}(x, z^{(k)})$
4:    Train auxiliary decoder $D_\phi^{(k)}$ on $\mathcal{B}$ using Equation A.1
5:    Compute SAI $\chi_{l,d,t}^{i,(k)}$
6:    Derive feedback instruction $\tilde{x}^{(k+1)}$ and generate feedback $c^{(k+1)} \sim f_\theta^{\mathrm{LLM}}(x, \tilde{x}^{(k+1)}, z^{(k)}, f_C^{(k)})$
7:    Update tokens $z^{(k+1)} \sim f_\theta^{\mathrm{LLM}}(x, c^{(k+1)})$
8: **end for**
9: Final protocol $f_C = (f_C^{(0)}, f_C^{(1)}, f_C^{(2)})$
10: **for** each training iteration do **do**
11:    **for** for each environment step $t$ **do**
12:       Obtain messages $m_t^i = f_C(\tau_t)$
13:       Encode latent $z_t^i = \mathrm{Enc}_\psi(\tau_t^i, m_t^i)$
14:       Compute joint loss $\mathcal{L}(\psi, \omega)$ using Equation A.5
15:       Update parameters $\psi, \omega$
16:    **end for**
17:    Periodically update target network $\omega^- \leftarrow \omega$
18: **end for**

---

objectives are defined as follows:

$$\mathcal{L}_{\mathrm{recon}}(\psi) = \mathbb{E}_{(\tau,s)\sim\mathcal{D}} \left[ \frac{1}{ND} \sum_{i=1}^{N} \sum_{d=1}^{D} \left( \|\hat{s}_{d,t}^i - s_{d,t}\|_2^2 + \lambda_{\mathrm{meta}}\, \mathrm{CE}(\hat{\chi}_{d,t}^i, \chi_{d,t}^i) \right) \right], \tag{A.2}$$

$$\mathcal{L}_{\mathrm{cons}}(\psi) = \mathbb{E}_{(\tau,s)\sim\mathcal{D}} \left[ \frac{1}{N} \sum_{i=1}^{N} \|\hat{z}_t^i - z_t^i\|_2^2 \right], \tag{A.3}$$

where CE denotes the cross-entropy loss, $\lambda_{\mathrm{meta}}$ controls the contribution of predicting $\chi_{d,t}^i$, and $\hat{z}_t^i$ is obtained by decoding and re-encoding $z_t^i$ through an auxiliary encoder $\mathrm{Enc}_{c,\psi}$ as $\hat{z}_t^i \sim \mathrm{Enc}_{c,\psi}(\mathrm{Dec}_\psi(z_t^i))$.

In parallel, we optimize the MARL policy using the TD loss used in QMIX, and jointly train policy learning and representation learning by minimizing the combined objective:

$$\mathcal{L}_{\mathrm{TD}}(\omega) = \mathbb{E}\left[ \left( r_t + \gamma \max_{a'} Q_{\omega^-}^{\mathrm{tot}}(s_{t+1}, a') - Q_\omega^{\mathrm{tot}}(s_t, a_t) \right)^2 \right]. \tag{A.4}$$

$$\mathcal{L}(\psi, \omega) = \mathcal{L}_{\mathrm{TD}}(\omega) + \mathcal{L}_{\mathrm{recon}}(\psi) + \lambda_{\mathrm{cons}}\, \mathcal{L}_{\mathrm{cons}}(\psi). \tag{A.5}$$

The overall training procedure of LMAC is summarized in Algorithm 1.

# B. Experimental Details

All baseline algorithms are evaluated using the official implementations and default settings released by their respective authors. The implementation of LMAC is based on EPyMARL[1] (Papoudakis et al., 2020), and all experiments in SMAC were conducted using StarCraft II version 2.4.10. Our method and comparisons are trained on an NVIDIA RTX 4090 GPU with an AMD EPYC 9334 CPU (Ubuntu 20.04). In the following sections, we provide details of the environments, baseline algorithms, reconstruction threshold settings, and hyper-parameter configurations used in our experiments. In particular, Appendix B.1 outlines the environment settings, Appendix B.2 describes the baseline algorithms, and Appendix B.3 summarizes the hyper-parameter configurations of our implementation.

## B.1. Environment Details

### B.1.1. STARCRAFT MULTI-AGENT CHALLENGE WITH COMMUNICATION

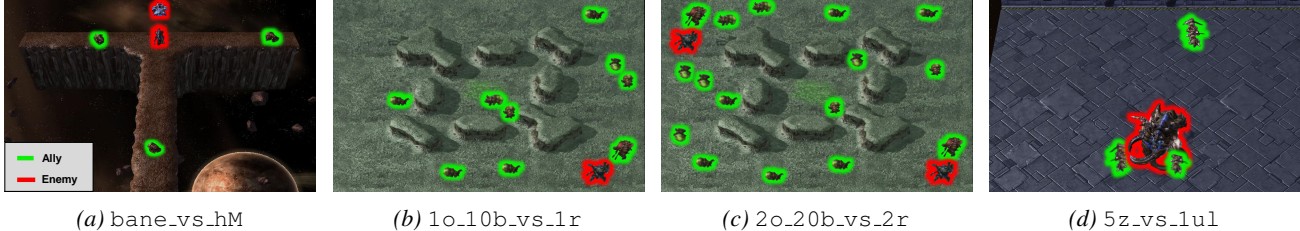

*(a)* `bane_vs_hM`    *(b)* `1o_10b_vs_1r`    *(c)* `2o_20b_vs_2r`    *(d)* `5z_vs_1ul`

*Figure B.1.* SMAC-Comm scenarios: (a) `bane_vs_hM`, (b) `1o_10b_vs_1r`, (c) `2o_20b_vs_2r`, (d) `5z_vs_1ul`.

We evaluate our method on four scenarios from the StarCraft Multi-Agent Challenge with Communication (SMAC-Comm). Among them, `bane_vs_hM`, `1o_10b_vs_1r`, and `5z_vs_1ul` are introduced by Wang et al. (2019), while `2o_20b_vs_2r` is a new map we propose based on `1o_10b_vs_1r`. The illustrations of these scenarios are shown in Fig. B.1, and their detailed configurations are summarized in Table B.1.

In SMAC-Comm, the **state space** contains absolute information of all units, including their positions, health, shields, energies, cooldowns, unit types, and most recent actions, while each agent's **observation space** is restricted to local information within its sight range, capturing relative positions, health, shield status, and unit types of nearby allies and enemies. The **action space** is defined as a set of discrete actions, including movement in four directions, attacks on visible enemies, special unit abilities, as well as stop and no-op commands, where no-op is used exclusively by eliminated units. The reward function is shaped by damage dealt to enemies, elimination of enemy units, and winning the scenario, and is formally defined as

$$R = \sum_{e \in \text{enemies}} \Delta\text{Health}(e) + \sum_{e \in \text{enemies}} \mathbb{I}(\text{Health}(e) = 0) \cdot \text{Reward}_{\text{death}} + \mathbb{I}(\text{win}) \cdot \text{Reward}_{\text{win}} \quad \text{(B.1)}$$

where $\Delta\text{Health}(e)$ denotes the health reduction of enemy unit $e$ during a timestep, $\mathbb{I}(\cdot)$ is an indicator function, and $\text{Reward}_{\text{death}}$ and $\text{Reward}_{\text{win}}$ are set to 10 and 200, respectively. A more detailed description of each scenario is provided below.

**bane_vs_hM:** Three Banelings attempt to take down a Hydralisk supported by a Medivac. Only when all three explode together can the Hydralisk be defeated, as any delay allows the Medivac to restore its health. To succeed, the Banelings must strike in perfect unison at the central junction of the T-shaped map, where the Hydralisk is positioned. This scenario requires agents to accurately perceive their positions and execute attacks simultaneously in order to succeed.

**1o_10b_vs_1r:** On a cliff-dense map, an Overseer locates a Roach that must be eliminated by its 10 Baneling allies to secure victory. While the Overseer and Roach appear together at a random spot, the Banelings spawn separately across the map. Under a minimal communication scheme, the Banelings remain silent, leaving the Overseer responsible for encoding its own position and transmitting it to guide the team.

**2o_20b_vs_2r:**. This map is an extension of `1o_10b_vs_1r` that we propose. Similar to the original setting, the scenario is played on a cliff-dense map where 20 Banelings must eliminate 2 Roaches. Both Roaches and Banelings spawn at random

---

[1]https://github.com/uoe-agents/epymarl

locations across the map. This environment is designed to evaluate whether our proposed communication method remains effective in more complex scenarios with a larger number of agents.

**5z_vs_1ul:** This map features five Zealots controlled by the agents against one Ultralisk as the enemy. The Ultralisk has high health and strong melee attacks, requiring coordinated micro-management from the Zealots to win. The challenge emphasizes some tactics such as kiting strategies, positioning, and focus fire to maximize damage while minimizing losses.

*Table B.1.* Detailed description of SMAC-Comm scenarios

| Map | Ally Units | Enemy Units | State Dimension | Obs Dimension | Num. of Actions |
|---|---|---|---|---|---|
| bane_vs_hM | 3 Banelings | 1 Hydralisk, 1 Medivac | 52 | 31 | 8 |
| 1o_10b_vs_1r | 1 Overseer, 10 Banelings | 1 Roach | 148 | 85 | 7 |
| 2o_20b_vs_2r | 2 Overseers, 20 Banelings | 2 Roaches | 296 | 171 | 7 |
| 5z_vs_1ul | 5 Zealots | 1 Ultralisk | 63 | 36 | 7 |

### B.1.2. SMACv2

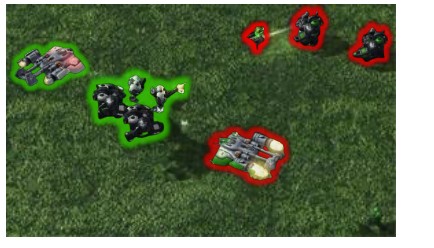
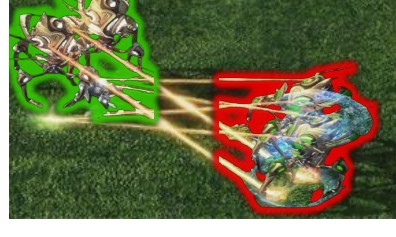
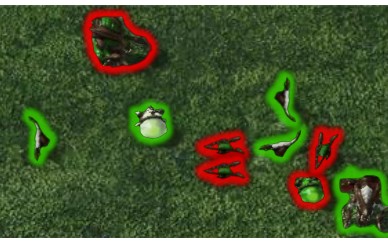

*(a)* terran_5_vs_5      *(b)* protoss_5_vs_5      *(c)* zerg_5_vs_5

*Figure B.2.* Visualizations of SMACv2 scenarios

SMACv2 (Ellis et al., 2023) extends the original SMAC benchmark (Samvelyan et al., 2019) to enable a more thorough evaluation of generalizable cooperative MARL. It largely preserves SMAC's state, observation, and action interfaces and its reward structure, but increases difficulty through stochasticity and diversity. Map configurations are no longer fixed: unit positions, health, counts, and attributes are randomized at initialization, which discourages memorization of static scenarios. The benchmark also features more diverse units and more asymmetric matchups, requiring richer tactical coordination. Moreover, difficulty scaling introduces additional randomness in environmental factors beyond enemy composition and placement, naturally creating a train–test distribution shift. In our experiments, we consider `terran_5_vs_5`, `protoss_5_vs_5` and `zerg_5_vs_5`. Visualizations are provided in Fig. B.2 and scenario details are summarized in Table B.2.

*Table B.2.* Detailed description of SMACv2 scenarios. Probabilities specify the sampling distribution over randomized compositions.

| Map | Ally Units (prob.) | Enemy Units (prob.) | State Dim | Obs Dim | Num. Actions |
|---|---|---|---|---|---|
| terran_5_vs_5 | Marine (0.45) Marauder (0.45) Medivac (0.1) | Marine (0.45) Marauder (0.45) Medivac (0.1) | 120 | 82 | 11 |
| zerg_5_vs_5 | Zergling (0.45) Hydralisk (0.45) Baneling (0.1) | Zergling (0.45) Hydralisk (0.45) Baneling(0.1) | 120 | 82 | 11 |
| protoss_5_vs_5 | Zealot (0.45) Stalker (0.45) Colossus (0.1) | Zealot (0.45) Stalker (0.45) Colossus (0.1) | 130 | 92 | 11 |

### B.1.3. LEVEL-BASED FORAGING

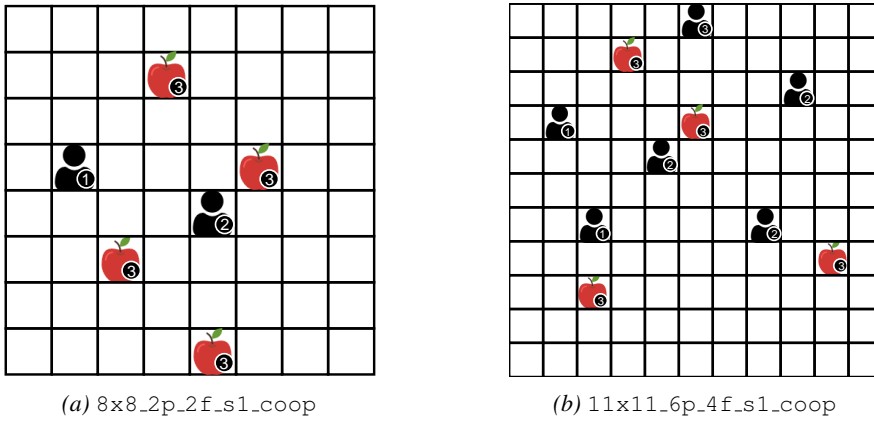

*(a)* `8x8_2p_2f_s1_coop`  *(b)* `11x11_6p_4f_s1_coop`

*Figure B.3.* LBF scenarios: (a) `8x8_2p_2f_s1_coop`, (b) `11x11_6p_4f_s1_coop`.

We adopt the Level-Based Foraging (LBF) variant introduced by Li et al. (2022b). The **state space** is represented as a structured grid encoding the positions and levels of all agents along with the locations and required levels of food items, rather than by concatenating individual observations. With the cooperation option enabled, each food item requires the joint effort of multiple agents, with its level set equal to the sum of the three lowest agent levels, ensuring that no agent can collect food alone and that every successful loading demands coordination. The **observation space** for each agent is limited to a $3 \times 3$ local field centered on itself, capturing relative information about nearby agents and food. The **action space** consists of six discrete actions: moving north, south, east, or west, attempting to load adjacent food, and the idle action (none). The **reward function** is cooperative and normalized by the total potential food value, granting positive returns only when the combined levels of participating agents meet or exceed the requirement of the targeted food. We evaluate two cooperative configurations as illustrated in Fig. B.3.

**8x8_2p_2f_s1_coop:** A compact $8 \times 8$ grid with 2 agents and 2 food items, where cooperation is strictly enforced for every collection attempt.

**11x11_6p_4f_s1_coop:** A larger $11 \times 11$ grid with 6 agents and 4 food items under the same cooperative setting, introducing greater complexity through increased agent interactions and map size.

### B.1.4. GOOGLE FOOTBALL RESEARCH

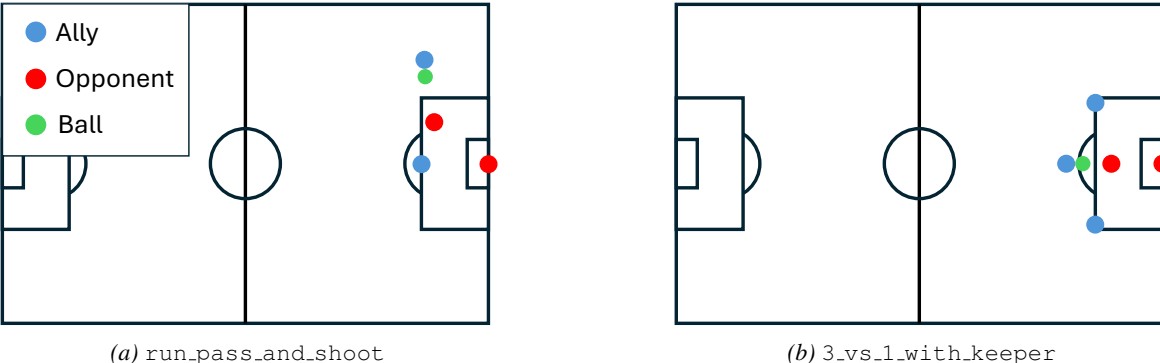

*(a)* `run_pass_and_shoot`  *(b)* `3_vs_1_with_keeper`

*Figure B.4.* GRF scenarios: (a) `Run_pass_and_shoot`, (b) `3_vs_1_with_keeper`.

We use the Google Research Football (GRF) environment (Kurach et al., 2020), a physics-based soccer simulator that incorporates core mechanics such as ball control, passing, shooting, tackling, and player movement. In this environment, each agent controls an individual player and must cooperate with teammates to score goals against scripted opponents. From the GRF scenarios, we consider `academy_3_vs_1_with_keeper` and `academy_run_pass_and_shoot_with_keeper`,

which we refer to as `3_vs_1_with_keeper` and `run_pass_and_shoot` for brevity. The illustrations of the GRF scenarios are shown in Figure B.4, and their detailed configurations are summarized in Table B.3.

In GRF, the **state space** contains the positions and velocities of all players as well as the ball, with ally and opponent features represented in the same format. Each agent's **observation space** consists of local information about itself, nearby teammates, opponents, and ball-related features, all expressed relative to the agent's frame. The **action space** is discrete and includes movement in eight directions, sliding, passing, shooting, sprinting, and standing still, which together enable the agents to create scoring opportunities. The **reward function** is provided under two schemes: Scoring and Checkpoint. The Scoring function gives +1 for scoring a goal and -1 for conceding, while the Checkpoint function provides additional intermediate rewards such as successful passes or defensive actions. In our experiments, we adopt the sparse Scoring function to increase the difficulty of the scenarios. A more detailed description of each scenario is provided below.

**3_vs_1_with_keeper:** Three attackers operate from the edge of the box: one on each wing and one in the center. The central player begins with the ball while directly confronted by a defender, and an opposing goalkeeper guards the net. The scenario emphasizes teamwork through passing and positioning to create scoring opportunities.

**run_pass_and_shoot:** Two attackers are positioned near the edge of the penalty area. One player starts wide with possession and is unmarked, while the other is placed centrally, marked by a defender, and facing the goalkeeper. The setup encourages passing and coordinated shooting to overcome the defense.

*Table B.3.* Detailed description of GRF scenarios

| Scenario | Ally | Opponent | State Dim | Obs Dim | Action Dim |
|---|---|---|---|---|---|
| `3_vs_1_with_keeper` | 3 central midfield | 1 goalkeeper, 1 center back | 26 | 26 | 19 |
| `Run_pass_and_shoot` | 2 central back | 1 goalkeeper, 1 center back | 22 | 22 | 19 |

## B.2. Detailed Description of Baseline Algorithms

**QMIX (Rashid et al., 2020)** QMIX factorizes the joint action-value function into individual utilities using a monotonic mixing network. It provides a strong baseline for cooperative MARL under centralized training with decentralized execution, but does not involve explicit communication between agents. We base our implementation of QMIX on the following repository: `https://github.com/hijkzzz/pymarl2`

**FullComm** A variant of QMIX where each agent broadcasts its full local observation to all others at every timestep. This represents an upper-bound setting with maximal communication capacity but incurs heavy redundancy and communication cost.

**QMIX+State** An oracle-like upper bound where each agent is directly given the global state in addition to its local observation. This allows agents to make fully informed decisions and serves as a reference for the maximum achievable performance.

**TarMAC (Das et al., 2019)** Targeted Multi-Agent Communication employs a signature-based soft attention mechanism that enables agents to actively select communication recipients. Senders broadcast a message comprising a signature key and a value, while receivers generate a query to compute attention weights via dot-product matching, allowing them to selectively integrate relevant information. Furthermore, the framework supports multi-round communication, enabling agents to coordinate through multiple stages of reasoning before taking action. We refer to the implementation released by the NDQ authors: `https://github.com/TonghanWang/NDQ`

**NDQ (Wang et al., 2019)** Neural Decomposable Q-learning introduces nearly decomposable Q-functions that minimize communication overhead. Agents act independently most of the time, but exchange messages guided by information-theoretic regularizers that maximize mutual information while minimizing entropy. This approach achieves strong coordination while reducing communication by over 80% compared to full exchange. The official code can be found at: `https://github.com/TonghanWang/NDQ`

**MASIA (Guan et al., 2022)** Multi-Agent Self-supervised Information Aggregation enables agents to aggregate received raw messages into compact, permutation-invariant representations. These embeddings are optimized through self-supervised objectives such as reconstruction and prediction, allowing agents to extract the most relevant information for decision-making and significantly improve coordination. The official code can be found at: `https://github.com/chenf-ai/Multi-Agent-Communication-Considering-Representation-Learning`

**MAIC (Yuan et al., 2022)** Multi-Agent Incentive Communication allows each agent to generate incentive messages that directly bias teammates' value functions, promoting explicit coordination. By learning targeted teammate models and applying sparsity regularization, MAIC improves efficiency and achieves strong performance across diverse cooperative MARL benchmarks. The official code can be found at: `https://github.com/mansicer/MAIC`

**SMS (Xue et al., 2022)** Efficient Multi-Agent Communication via Shapley Message Value reduces redundancy by modeling communication as a cooperative game and using the Shapley Message Value (SMV) to quantify each teammate message's marginal utility. A Shapley Message Selector predicts SMVs from local observations, so agents request only messages with positive predicted value, enabling efficient targeted communication. Code: `https://github.com/DiXue98/SMS`

**COLA (Xu et al., 2023)** Consensus Learning for Agents enables cooperative behavior by allowing agents to infer a shared consensus representation from their local observations. Even without direct access to the global state, agents learn viewpoint-invariant representations that converge to the same discrete consensus, which is then used as an additional input for decentralized decision-making. The official code can be found at: `https://github.com/deligentfool/COLA`

**T2MAC (Sun et al., 2024)** Targeted and Trusted Multi-Agent Communication equips agents with mechanisms for selective engagement and evidence-driven message integration. Agents decide when and with whom to communicate, exchange individualized messages, and integrate received information at the evidence level, leading to more efficient and reliable cooperation. The official code can be found at: `https://github.com/ZangZehua/T2MAC`

### B.3. Hyper-parameter Setup

We determined the reconstruction threshold $\alpha$ by aligning the statistical distribution of errors with the physical spatial semantics of each environment. Specifically, initial thresholds were derived to capture meaningful deviations relative to the actual state structure (e.g., unit radius or grid cell size) and were subsequently refined through empirical validation to ensure discriminative feedback. Following this calibration process, we set $\alpha$ to 0.05 for SMAC-Comm, where normalized coordinates directly reflect spatial error, and a stricter 0.005 for the challenging `bane_vs_hM` scenario. For GRF, we adopted 0.002 given its absolute field coordinates and weaker observability limits, while for the grid-based LBF, we set $\alpha$ to 0.1 to account for cell occupancy prediction. Beyond these thresholds, default hyper-parameters were used as the baseline configuration. For each scenario, we primarily followed the settings provided by the original authors; when such specifications were unavailable, the default parameters were applied. The full set of hyper-parameters used in our experiments is summarized in Table B.4.

*Table B.4.* Common hyper-parameter setting of LMAC

| Parameter | Value |
|---|---|
| Hidden dimension for self-attention module | 64 |
| Latent dimension | 20 |
| Dropout rate | 0.1 |
| Optimizer | Adam |
| $\epsilon$ anneal step | 50000 |
| $\epsilon$ Decay Value | $1.0 \rightarrow 0.05$ |
| Replay buffer size | 5000 |
| Target update interval | 200 |
| Mini-batch size | 32 |
| Mixing network dimension | 32 |
| Discount factor $\gamma$ | 0.99 |
| Learning rate | 0.0005 |
| Coefficient of meta-information loss $\lambda_{\text{meta}}$ | 0.1 |
| Coefficient of consistency loss $\lambda_{\text{cons}}$ | 1 |
| Temperature (LLM generation) | 0.6 |

## C. Additional Trajectory Analysis

In addition to the main trajectory analysis presented in the paper, we further examine protocol refinement in other SMAC-Comm scenarios and GRF tasks. These supplementary cases demonstrate that the framework yields step-wise refined communication protocols whose iterative refinements adapt to scenario-specific challenges. The following analyses provide examples from different environments, illustrating how the protocol evolves beyond the scenarios presented in Section 5.2.

**SMAC-Comm: bane_vs_hM**

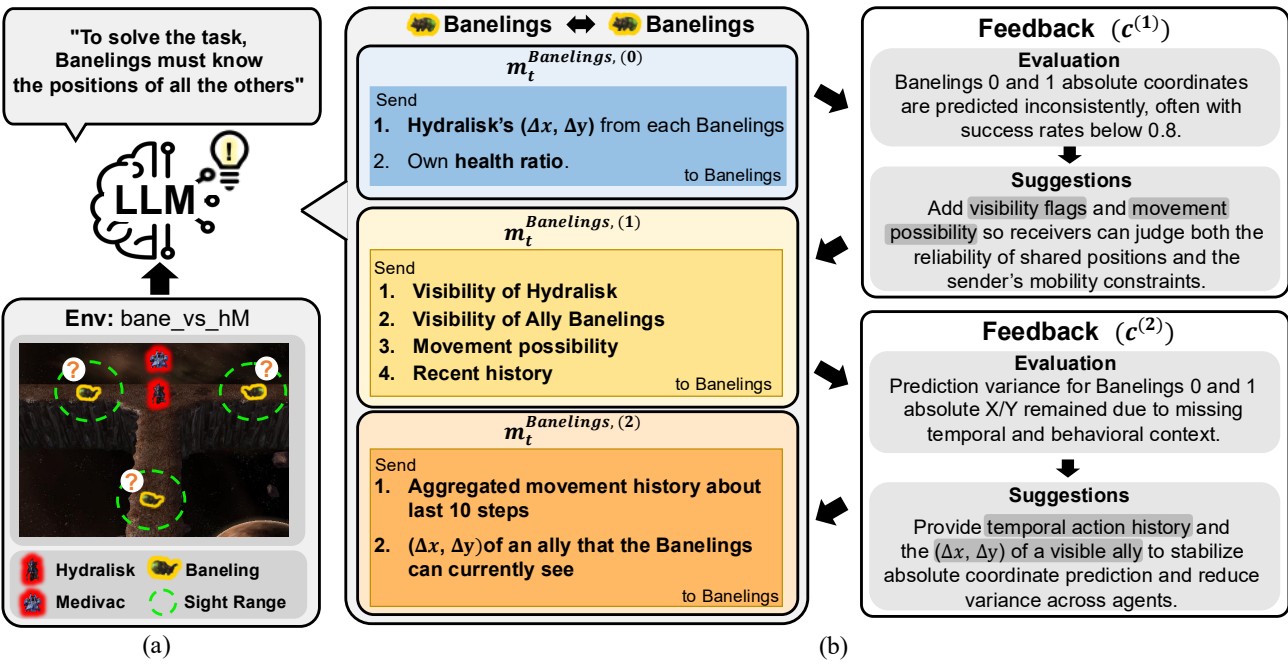

*Figure C.1.* Protocol refinement analysis on SMAC bane_vs_hM: (a) Task scenario with Banelings, a Hydralisk, and a Medivac under partial observability, (b) protocol messages and corresponding feedback at each step $k$

As a complementary case, we analyze protocol refinement in the SMAC bane_vs_hM map, summarized in Fig. C.1. In (a), three Banelings must coordinate a near-simultaneous detonation against a Hydralisk supported by a Medivac, where tight synchronization is critical. Under partial observability, local observations do not provide absolute coordinates, and the long vertical corridor makes inferring the $y$-position particularly challenging, which often leads to misaligned engagement timing without additional localization cues. In (b), we illustrate the step-wise protocol refinement. At $k = 0$, Banelings share the Hydralisk's relative offset $(\Delta x, \Delta y)$ and their health ratio, which provides only partial localization and yields unstable absolute state recovery across agents. Criterion-based feedback derived from recovery statistics indicates that this instability stems from missing information about visibility and feasible motion, motivating the inclusion of visibility indicators, movement availability, and short-term history. At $k = 1$, these additions increase recovery success rates, yet inter-agent variance remains high for specific coordinates due to insufficient temporal and behavioral context. At $k = 2$, variance-based feedback further refines the protocol by adding aggregated movement history over the last 10 steps along with the relative positions of currently visible allies, which stabilizes absolute recovery and reduces inter-agent discrepancies. Taken together, this case shows that protocol refinement guided by recovery and imbalance criteria promotes consistent absolute localization and reliable synchronization by introducing structured temporal-behavioral cues beyond instantaneous relative observations.

**GRF: Run_pass_and_shoot**

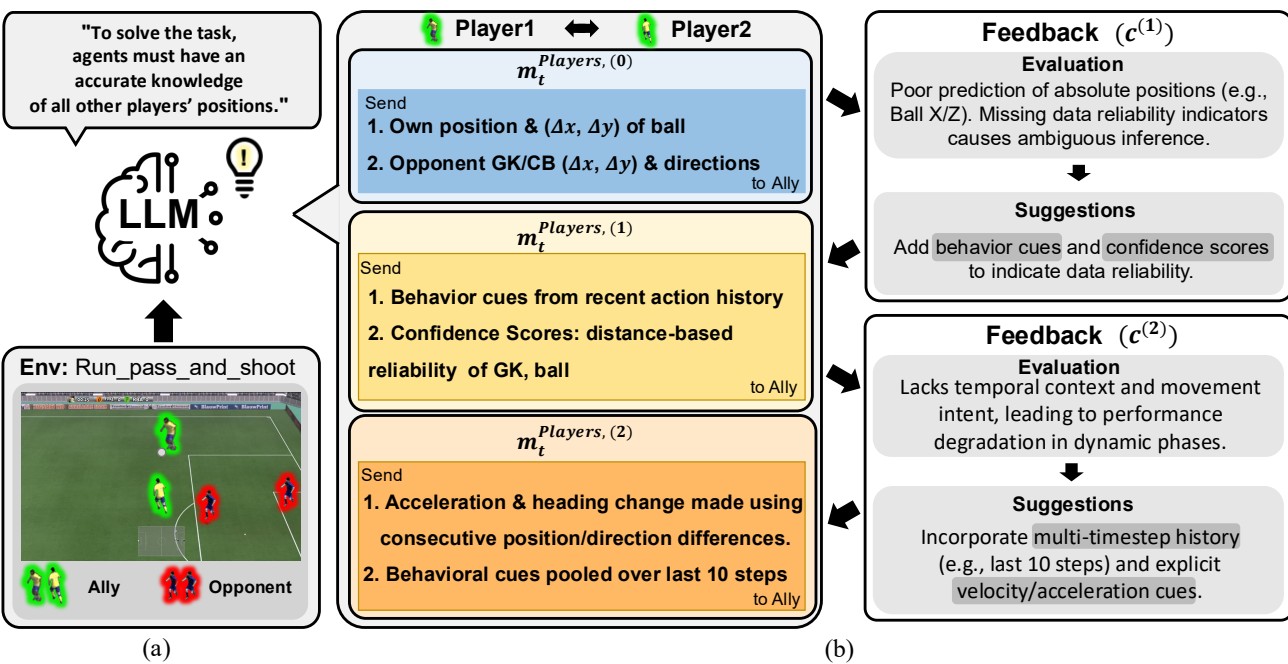

*Figure C.2.* Protocol refinement analysis on GRF `Run_pass_and_shoot`: (a) Task scenario with two attackers, a defender, and goalkeeper near the penalty area, (b) protocol messages and corresponding feedback at each step $k$

As a complementary case, we analyze protocol refinement in the GRF `Run_pass_and_shoot` scenario, summarized in Fig. C.2. In (a), two attackers must cooperate near the penalty area against a central defender and a goalkeeper. Although the state space in GRF is structurally simpler than in SMAC, it remains important that agents infer states from shared messages and incorporate them into policy decisions, a pattern that is also observed in LBF. (b) shows the protocol evolution. At $k = 0$, each agent shares its own position, the relative displacement of the ball, and the positions of the central defender and goalkeeper, but such information alone is limited for predicting other aspects of the state. Accordingly, at $k = 1$, behavioral cues such as pass/shoot readiness and sprinting, together with confidence scores regarding the goalkeeper and ball, are added. At $k = 2$, dynamic features such as acceleration and heading changes, along with aggregated behavioral histories over the last 10 steps, are incorporated, stabilizing predictions and enabling cooperative play in which the wide attacker penetrates open space while the central striker draws defensive pressure.

# D. Additional Experimental Analyses

In this section, we present additional experimental analyses to provide deeper insights into LMAC. We first investigate the effect of iterative protocol refinement in Appendix D.1. We then compare LLM-designed messages with message designs from baseline methods in Appendix D.2, and examine robustness to the initial offline dataset and to moderate environment changes in Appendix D.3. Next, we assess robustness under communication constraints by limiting message dimensions in Appendix D.4. Finally, we analyze computational complexity and token usage in Appendix D.5, showing that LMAC adds only modest overhead.

## D.1. Effect of the Number of Refinement Iterations

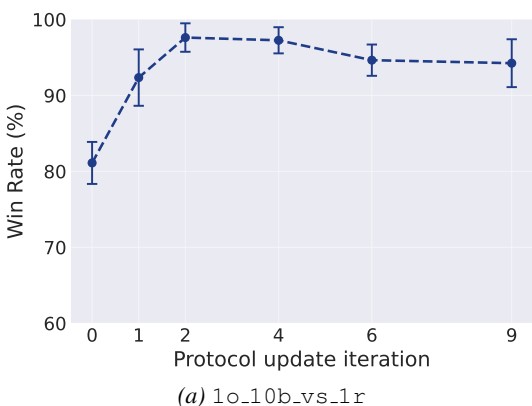
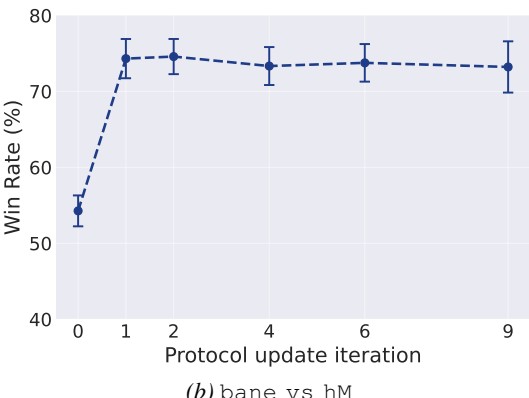

*(a)* `1o_10b_vs_1r`       *(b)* `bane_vs_hM`

*Figure D.1.* Performance across the number of update steps in (a) `1o_10b_vs_1r` and (b) `bane_vs_hM`.

We analyze the effect of repeatedly applying feedback-based protocol refinement using the Sharing Enhancement update scheme. As shown in Fig. D.1, performance improves as the number of update iterations $k$ increases, but the marginal gain quickly saturates around $k = 3$. In fact, even $k = 2$ is sufficient to capture most important state dimensions that can be inferred from observations, while larger $k$ mainly increases message size and introduces redundant information, reducing efficiency. Nevertheless, in environments that demand more sophisticated reasoning, employing more refinement steps may still offer benefits.

## D.2. Comparison with Alternative Message Designs

*Table D.1.* Comparison of LMAC variants under different message designs (win rate, %).

| Method | 1o_10b_vs_1r | bane_vs_hM |
|---|---|---|
| LMAC | $96.2 \pm 1.6$ | $75.2 \pm 3.5$ |
| LMAC (w/ SMS message) | $59.0 \pm 4.5$ | $23.5 \pm 4.3$ |
| LMAC (w/ TarMAC message) | $25.2 \pm 2.7$ | $16.6 \pm 4.5$ |
| SMS | $56.1 \pm 3.8$ | $21.5 \pm 4.2$ |
| TarMAC | $22.5 \pm 4.1$ | $12.1 \pm 7.2$ |

While the ablation study in Section 5.1 shows that the refinement step itself is meaningful, it is also important to isolate the effect of the LLM reasoning used to design the communication protocol. To this end, we compare LMAC with variants that replace the LLM-designed message with messages from other gradient-based communication methods, specifically SMS (Xue et al., 2022) and TarMAC (Das et al., 2019). We denote these variants as *LMAC (w/ SMS message)* and *LMAC (w/ TarMAC message)*. As shown in Table D.1, LMAC achieves the best performance among all variants in both scenarios. This indicates that the overall gain is not explained solely by the downstream training pipeline, and that LLM reasoning in LMAC contributes substantially to the effectiveness of the learned communication protocol.

## D.3. Robustness to Offline Dataset and Environment Changes

*Table D.2.* Different initial offline datasets.

| Dataset | 1o_10b_vs_1r | bane_vs_hM |
|---------|---------------|-------------|
| Original | $96.2 \pm 1.6$ | $75.2 \pm 3.5$ |
| Random | $95.7 \pm 1.8$ | $74.3 \pm 3.6$ |

*Table D.3.* Modified settings on `bane_vs_hM`.

| Setting | Win rate (%) |
|---------|--------------|
| Original | $75.2 \pm 3.5$ |
| Spawn switch | $73.4 \pm 3.9$ |
| Visibility reduction | $74.2 \pm 4.1$ |

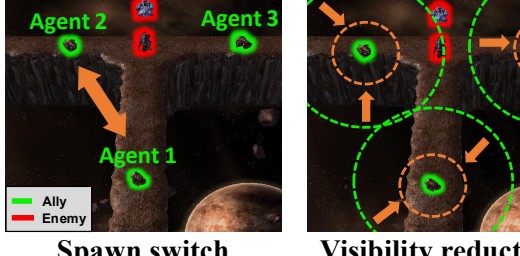

**Spawn switch**          **Visibility reduction**

*Figure D.2.* Illustration of the spawn switch and visibility reduction settings.

**Initial Offline Dataset.** To examine whether LMAC depends strongly on the quality of the initial offline dataset, we additionally construct $\mathcal{B}$ using trajectories collected from a random policy instead of during QMIX training. As shown in Table D.2, the performance difference is small in both scenarios. This suggests that LMAC is not highly sensitive to the quality of the initial dataset, likely because its protocol refinement is guided by how well messages support state recovery rather than by imitating a strong policy.

**Moderate Environment Changes.** We further examine whether the learned protocol remains effective when the environment is moderately changed. Specifically, the protocol is learned in the original `bane_vs_hM` setting, and then evaluated under two modified settings: one with switched spawn positions and another with reduced visibility. Fig. D.2 provides an illustration of these two modified settings. As shown in Table D.3, performance decreases only slightly under both modifications while remaining far above QMIX. This indicates that the learned protocol is not tied to a single fixed layout, but continues to provide useful information sharing under moderate changes in the environment.

## D.4. Effect of Reduced Communication Capacity under Constraints

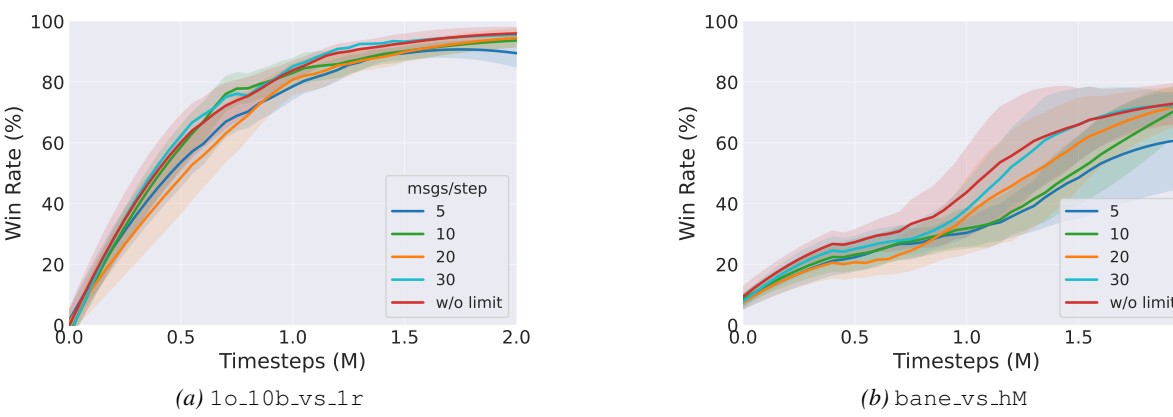

*(a)* `1o_10b_vs_1r`          *(b)* `bane_vs_hM`

*Figure D.3.* Performance comparison under message dimension constraints in (a) `1o_10b_vs_1r` and (b) `bane_vs_hM`.

We investigated how performance changes when message dimensionality is constrained, which effectively limits communication capacity. Here, we use communication capacity to denote the effective amount of information that agents can transmit through messages, which is directly determined by message dimensionality. Thus, restricting the number of message dimensions can be regarded as limiting the communication capacity of agents. As shown in Fig. D.3, our method remains robust, largely preserving performance even with reduced message sizes. The LLM minimizes overhead by designing protocols that utilize one-way broadcasts or compactly encode key features like movement and last action into fewer bits. However, in more challenging scenarios such as `bane_vs_hM`, where state inference is inherently more difficult, excessive compression slowed convergence, indicating that a moderate level of communication capacity is still necessary for effective learning.

## D.5. Comparison of Computational Complexity & Token Analysis

**Computational Complexity.** In the SMAC-Comm experiments, we report the total wall-clock training time for 2M steps on the `bane_vs_hM` and `1o_10b_vs_1r` maps in Table D.4. Overall, LMAC incurs a modest runtime overhead: on average, it requires about $6\%$ more training time than strong baselines such as MASIA and MAIC. This overhead mainly stems from optimizing the auxiliary decoder, which diagnoses weaknesses in the current communication protocol and supports step-wise refinement. Consistently, Table D.5 shows that auxiliary decoder training dominates the additional runtime, taking about 27 minutes on average per run during protocol initialization and recovery enhancement, indicating that the primary cost is attributable to protocol diagnosis and improvement rather than the base MARL learner.

*Table D.4.* Total training time (hours) for 2M steps in SMAC communication settings.

| Algorithm | `bane_vs_hM` | `1o_10b_vs_1r` |
|---|---|---|
| QMIX | 4h 12m | 5h 42m |
| NDQ | 5h 17m | 6h 33m |
| T2MAC | 5h 34m | 7h 28m |
| MASIA | 7h 13m | 8h 13m |
| MAIC | 7h 45m | 8h 32m |
| COLA | 7h 54m | 8h 24m |
| LMAC(Ours) | 7h 56m | 8h 47m |

*Table D.5.* Runtime analysis of LLM inference and auxiliary decoder training.

| | Time (min) | | | | |
|---|---|---|---|---|---|
| Component | Baseline | $k = 0$ | $k = 1$ | $k = 2$ | Total |
| LLM inference | – | 0.45 | 0.65 | 0.65 | 1.75 |
| Auxiliary decoder training | 8.94 | 9.00 | 9.34 | – | 27.28 |
| **Total** | **8.94** | **9.45** | **9.99** | **0.65** | **29.03** |

**Token Consumption and LLM Cost.** Because complex MARL requires many environment interactions, *in-the-loop* methods that call an LLM at every timestep can make API cost and latency major bottlenecks. In contrast, LMAC never invokes the LLM during online execution and refines the protocol offline in only a few steps using criteria from RL transition data. As shown in Table D.6, LMAC uses 70.4k tokens per run, consisting of 59.6k input tokens and 11.0k output tokens, which corresponds to an estimated GPT-4.1 API cost of $0.227 per run. The average retry count is also low at 2.25, indicating stable protocol and feedback generation. Table D.5 further shows that total LLM inference takes only 1.75 minutes, which is substantially smaller than the auxiliary decoder training time of 27.28 minutes, indicating that LLM usage contributes only a minor and predictable overhead.

*Table D.6.* Sequential token consumption, retry rates, and GPT-4.1 API cost (USD).

| Sequence iteration | Input (k) | Output (k) | Total (k) | Avg. Retries | Cost (USD) |
|---|---|---|---|---|---|
| **Protocol Initialization** ($k = 0$) | 6.3 | 2.8 | 9.1 | 0.75 | 0.034 |
| **Feedback Generation. 1** | 10.9 | 0.8 | 11.6 | 0.00 | 0.028 |
| **Recovery enhancement** ($k = 1$) | 9.4 | 3.1 | 12.5 | 0.50 | 0.044 |
| **Feedback Generation. 2** | 13.3 | 0.8 | 14.0 | 0.00 | 0.033 |
| **Imbalance mitigation** ($k = 2$) | 19.7 | 3.5 | 23.2 | 1.00 | 0.067 |
| **Total per Run** | **59.6** | **11.0** | **70.4** | **2.25** | **0.227** |

