# OpenReview forum: "LLM-Guided Communication for Cooperative Multi-Agent Reinforcement Learning"
_ICML.cc/2026/Conference — ICML 2026 regular_

### Official Review · Reviewer_USMG · 2026-03-08

**Soundness:** 3
**Presentation:** 3
**Significance:** 3
**Originality:** 3
**Overall Recommendation:** 4
**Confidence:** 2

**Summary:**

This paper proposes LMAC, a framework that uses LLMs to design and iteratively refine executable, agent-wise communication protocols for cooperative MARL under partial observability. The final protocol is integrated into a CTDE learner with a meta-cognitive latent module, producing strong gains over communication baselines on SMAC-Comm, LBF, and GRF.

**Compliance With Llm Reviewing Policy:**

Affirmed.

**Final Justification:**

I think the authors have addressed my concerns, as well as other reviewers' concerns. Therefore, I keep the score of weak accept.

**Key Questions For Authors:**

1.	The core of this paper is that an LLM can design better communication protocols by reasoning about task descriptions and observation dimensions. My question is: how to determine whether the gains are due to LLM reasoning or the additional design stage itself. I think there is need to compare LLM-designed communication with other communication protocols such as gradient-based optimization, or similar protocols using representation-learning approaches.
2.	The real-world MARL environments require adaptive communication policies that evolve as the agents learn. I’m wondering whether the protocol can remain optimal as the agents’ policies change during training.
3.	The best-performing alpha per task is selected. Can you report performance as a function of alpha to demonstrate robustness?
4.	How many LLM calls and tokens are used per scenario for the full two-step refinement? What fraction of total training wall-clock do the LLM/decoder stages occupy?
5.	The minimal communication required for state reconstruction is emphasized in the paper, but there is no theoretical description provided. Please explain more about this.

**Limitations:**

Yes.

**Strengths And Weaknesses:**

Strength:
This paper proposes an original use of LLMs to synthesize executable, agent-wise communication code from structured task/state/observation descriptions, avoiding online LLM calls at execution time.
This paper introduces a principled, criterion-based, two-step refinement loop that grounds LLM feedback entirely in signals computed from RL transition data.
This paper valuates across three representative communication-heavy benchmarks, with multiple scenarios and five seeds, reporting both learning curves and state-recovery metrics.

Weakness:
The proposed approach assumes availability of accurate natural-language mappings for task/state/observation dimensions, which may not be readily available in many domains.
The two-step refinement is justified empirically, but the stopping rule is heuristic.
The convergence/stability guarantees of the refinement loop are not analyzed.

---

> ### Author Rebuttal · Authors · 2026-03-31
>
> We sincerely thank the reviewer for the thoughtful and in-depth questions. We would like to address the reviewer’s concerns as follows. We will incorporate all of the points addressed in our response into the revised manuscript.
>
> ---
> **Weakness 1 (Dependence on input prompt design).** We agree that this is an important issue. Since similar concerns were also raised by other reviewers, we evaluate simplified versions of both $\mathcal{I}_T$ and $\mathcal{I}_P$. By the reviewer guidelines, we kindly refer the reviewer to our response to Reviewer gWcq (Table R3). The results show that **LMAC remains effective under these simplified prompts**, indicating that **it is not highly sensitive to the input prompt design.** While some minimal prompt specification may still be needed for a new environment, we believe this remains well within a practical range.
>
> **Weakness 1 (Refinement step analysis).** As the reviewer pointed out, our work does not provide a convergence analysis for the refinement step, but the design is not purely heuristic. As analyzed in Section 5.2 and Appendix E.1, the two-step design already reduces the reconstruction error sufficiently to recover useful state information, while adding more steps mainly increases the message dimension without clear performance gains. This is why we set $K = 2$. We will clarify the rational in the manuscript.
>
> ---
> **Question 1 (Ablation on LLM reasoning).** We appreciate the reviewer’s important question. While the ablation study in Section 5.1 shows that the refinement step is meaningful, it is also important to verify the effect of **the LLM reasoning itself**. To isolate this, we compare LMAC with variants that replace the LLM-designed message with messages from other gradient-based methods, SMS and TarMAC. We denote these variants as **LMAC (w/ SMS message)** and **LMAC (w/ TarMAC message)**, and report the results in Table R4. The results show that LMAC achieves the best performance among all variants, indicating that LLM reasoning in LMAC contributes substantially to the overall performance gain.
>
> **Table R4.** Comparison of LMAC variants under different message designs (Win rate).
> |**Exp.**|**1o_10b_vs_1r**|**bane_vs_hM**|
> |---|---|---|
> |LMAC|$\mathbf{96.2\pm1.6}$|$\mathbf{75.2\pm3.5}$|
> |LMAC (w SMS message)|$59.0\pm4.5$|$23.5\pm4.3$|
> |LMAC (w TarMAC message)|$25.2\pm2.7$|$16.6\pm4.5$|
> |SMS|$56.1\pm3.8$|$21.5\pm4.2$|
> |TarMAC|$22.5\pm4.1$|$12.1\pm7.2$|
>
> ---
> **Question 2 (Validity under small environment changes).** We agree that validating robustness under environment changes is important. Following a similar concern raised by Reviewer sokR, we conduct additional experiments on bane_vs_hM with modified spawn positions and visibility. Due to space limitations, we refer the reviewer to Table R2 in our response to Reviewer sokR. The results show a slight performance drop, but the protocol remains effective and continues to provide meaningful gains. We attribute this to the fact that **LMAC is not tied to fixed positions or visibility**, but is designed to compensate for missing information under partial observability.
>
> ---
> **Question 3 (Robustness to threshold $\alpha$).** We study the effect of varying the threshold $\alpha$ on the SMAC-Comm environments in Table 1(c) of Section 5.3. Since $\alpha$ determines whether reconstruction is regarded as successful, performance can drop when it is set too small or too large. However, the drop is generally modest, suggesting that **the method is reasonably robust to changes in $\alpha$.** As additionally reported in Appendix B.3, for most environments, setting $\alpha$ in proportion to the state scale yields the best performance. Only for tasks such as bane_vs_hM, where accurate state reconstruction is particularly important, we use a smaller $\alpha$ to apply a stricter success criterion.
>
> ---
> **Question 4 (LLM token usage and overhead).** In Appendix E.3, we provide the token usage and complexity analysis. Briefly, for the 1o_10b_vs_1r scenario, the refinement steps requires approximately 70.4k tokens in total, corresponding to about $0.227 with the GPT-4.1 API. The total time for the protocol-design stage is about 29 minutes, which accounts for only 5.5% of the total MARL training time for that scenario. Therefore, **the additional cost remains practical in terms of both token usage and computational overhead.**
>
> ---
> **Question 5 (Minimal communication).** In our paper, the term **“minimal communication”** does not refer to an information-theoretically optimal code. Rather, it means that the protocol starts from a basic message and adds only the information necessary at each refinement step, until the reconstruction quality improves sufficiently. We agree that this wording may be open to misinterpretation, and will clarify it more explicitly in the revised manuscript.
>
> ---
> We once again thank the reviewer for the thoughtful feedback, and we hope that our responses have adequately addressed the reviewer’s concerns.

---

> > ### Author Rebuttal · Reviewer_USMG · 2026-04-01
> >
> > Thank you for your detailed response. My comments have been addressed. And I'll keep my original decision to accept this paper.

---

> > > ### Author Response · Authors · 2026-04-04
> > >
> > > We thank the reviewer for the important feedback and for maintaining a positive assessment of our work. The experiments and questions were highly helpful, and we will make sure to reflect them in the revised version.

---

### Official Review · Reviewer_gWcq · 2026-03-09

**Soundness:** 2
**Presentation:** 3
**Significance:** 3
**Originality:** 3
**Overall Recommendation:** 4
**Confidence:** 3

**Summary:**

The paper studies communication design for cooperative multi-agent reinforcement learning under partial observability. The motivation is that existing communication methods often either exchange information inefficiently or fail to provide enough information for agents to accurately recover the underlying state, which hurts coordination during decentralized execution. To address this, the paper proposes LMAC, an LLM-guided communication framework that uses an LLM to generate and iteratively refine an executable communication protocol so that agents can recover task-relevant state information more accurately and more uniformly. The learned protocol is then integrated into a CTDE MARL framework through a latent representation module. The method is tested on several cooperative MARL benchmarks, including SMAC-Comm, Level-Based Foraging, Google Research Football, and additional experiments on SMACv2.

**Compliance With Llm Reviewing Policy:**

Affirmed.

**Key Questions For Authors:**

- The method appears to rely heavily on carefully engineered prompts. How sensitive is performance to the exact prompt design, and do the gains remain under simpler or slightly varied prompts?

- LMAC seems to receive extra structured prior knowledge through natural-language descriptions of state and observation dimensions, unlike the baselines. Can the authors justify the fairness of this comparison, and can they rule out that this additional semantic information is a major source of the improvement?

**Limitations:**

yes

**Strengths And Weaknesses:**

Strengths

- The paper is well motivated, as it addresses an important challenge posed by partial observability in cooperative MARL.

- The paper is clearly written, and the overall method is presented in a well-organized and easy-to-follow manner.

- The paper is evaluated on a range of MARL tasks, including multiple cooperative benchmarks, which helps demonstrate the empirical effectiveness of the proposed approach across different environments.

Weaknesses

- The prompt design appears to be heavily hand-engineered. The method relies on carefully constructed task descriptions, protocol design instructions, step-specific feedback prompts, and structured output requirements.

- The baseline comparison does not seem entirely fair, since the proposed method is given additional structured prior knowledge through the prompt and through natural language descriptions of the state and observation, whereas the standard MARL communication baselines do not receive comparable semantic assistance.

---

> ### Author Rebuttal · Authors · 2026-03-31
>
> We sincerely thank the reviewer for the thoughtful and in-depth questions. We would like to address the reviewer’s concerns as follows. We will incorporate all of the points addressed in our response into the revised manuscript.
>
> ---
>
> **Weakness 1/Question 1 (Dependence on input prompt design).**
>
> We appreciate the reviewer’s concern and agree that this is an important issue. Since several reviewers raised a similar question, we provide additional experiments that simplify both $\mathcal{I}_T$ and $\mathcal{I}_P$. Specifically, for the task description $\mathcal{I}_T$, we consider a simplified version that uses a single sentence for the global state description: “The global state contains information about the status and spatial configuration of allied and enemy units, including attributes such as health, position, shield, unit type, and weapon cooldown. All values are normalized to [0,1].” For the observation description, instead of providing dimension-wise explanations, we use a coarser chunk-level description, specifying only the dimension ranges corresponding to agent-related features (ally/enemy) and recent-action features. For the protocol design instruction $\mathcal{I}_P$, we also simplify the prompt by keeping only the core communication objective, namely information sharing among agents, and the required output format.
>
> Based on this setup, we evaluate the following four variants:
>
> - Original LMAC
> - **V2:** LMAC with simplified $\mathcal{I}_T$ and full $\mathcal{I}_P$
> - **V3:** LMAC with full $\mathcal{I}_T$ and simplified $\mathcal{I}_P$
> - **V4:** LMAC with simplified $\mathcal{I}_T$ and simplified $\mathcal{I}_P$
>
> We report the comparison results in Table R3. The results show that **LMAC still operates successfully under these simplified prompts**, indicating that **its performance is not highly sensitive to the input prompt design.** While some minimal prompt specification may still be required for a new environment, we believe this remains well within a practical range.
>
>
> **Table R3.** Comparison of LMAC variants under simplified input prompts (Win rate).
>
> | **Exp.** | **1o_10b_vs_1r** | **bane_vs_hM** |
> | --- | --- | --- |
> | LMAC | $\mathbf{96.2 \pm 1.6}$ | $\mathbf{75.2 \pm 3.5}$ |
> | LMAC (V2) | $93.9 \pm 3.3$ | $72.0 \pm 3.2$ |
> | LMAC (V3) | $92.1 \pm 2.5$ | $72.5 \pm 3.7$ |
> | LMAC (V4) | $91.8 \pm 3.7$ | $71.5 \pm 3.6$ |
> | QMIX | $40.2 \pm 7.4$ | $19.5 \pm 2.1$ |
>
> ---
>
> **Weakness 2/Question 2 (Use of prior knowledge).**
>
> We agree with the reviewer that some degree of prior knowledge is helpful for the LLM to understand the environment. However, **this knowledge is used only for the LLM’s reasoning about what information should be exchanged, and is not used in the MARL training itself.** In this sense, we believe the overall comparison remains as fair as possible. Moreover, as shown in Table R3, which we include following the reviewer’s suggestion, substantially reducing such prior knowledge does not lead to a major drop in protocol performance. This suggests that a more minimal form of prior knowledge can also be a viable option. We thank the reviewer for this insightful comment.
>
> ---
>
> We once again thank the reviewer for the thoughtful feedback, and we hope that our responses have adequately addressed the reviewer’s concerns.

---

> > ### Author Rebuttal · Reviewer_gWcq · 2026-04-02
> >
> > Thank you for the rebuttal. My concerns have been adequately addressed, and I will keep my score as Weak Accept.

---

> > > ### Author Response · Authors · 2026-04-04
> > >
> > > We sincerely thank the reviewer for the thoughtful feedback and for maintaining a positive assessment of our work. The questions raised are highly valuable for improving the quality of the paper, and we will incorporate them into the revised manuscript.

---

### Official Review · Reviewer_sokR · 2026-03-12

**Soundness:** 2
**Presentation:** 3
**Significance:** 2
**Originality:** 3
**Overall Recommendation:** 4
**Confidence:** 3

**Summary:**

This paper present LMAC framework to address the two questions in MARL : inefficient method of information exchange and inability to convey sufficient status information. The core idea of LMAC is to use a large language model as an offline protocol designer that creates and iteratively refines how agents communicate with one another. LMAC improves the communication protocol generated from LLM in three refinement stages (protocol initiatlization, recovery enhancement and imbalance mitigation) using feedback derived from offline RL transition data. Experimentally, the paper tests LMAC on several benchmarks, which consistently outperforms a range of communication baselines.

**Compliance With Llm Reviewing Policy:**

Affirmed.

**Key Questions For Authors:**

1. It's not clear how much the method's success depends on the quality and detail of the semantic descriptions. We don't know if the improvements are due to the prepared semantic interface or just the way the optimization works.

2. How well does a protocol made for one specific setting work when the environment changes? This method seems to do well on several standard problems, but authors haven't tested how it handles small changes in space or structure without needing a complete redo.

3. What is the degree of dependence of this method on manually specified protocol constraints? In particular, how important is the requirement that communication codes be explicit, human-interpretable, and non-parametric (i.e., fixed and unlearnable)?

**Limitations:**

Yes.

**Strengths And Weaknesses:**

Strengths:
1. The refinement process is relatively interpretable rather than purely black-box. The feedback is structured in a step-specific manner, and the LLM is provided with both the previous protocol code and criterion-based performance signals, enabling it to revise concrete protocol logic instead of merely generating latent messages. It makes the method much easier to analyze than standard end-to-end communication learning.

2. The communication design principles are well justified. The protocol is built around state reconstruction, knowledge gap, uniqueness, sufficiency, and compactness, while also emphasizing explicit and efficient message design. It helps avoid the common failure mode of transmitting large but redundant messages

3. The evaluation is reasonably broad. It strengthens the empirical case for generality. It covers multiple benchmark families, includes a more challenging benchmark (SMACv2), examines different CTDE backbone.


Weaknesses:
1. This method seems to depend a lot on having detailed semantic descriptions of the surroundings ready. The prompt construction requires not only task descriptions, but also natural-language explanations of both state and observation dimensions. In more complex or less structured environments, this semantic annotation burden may become costly, unstable, and difficult to scale.

2. The method is complex. The paper argues that the LLM cost is offline, but the overall system complexity is still materially higher than standard MARL communication methods.

3. This approach doesn't automatically figure out the communication protocol from the initial data. Instead, it changes the task, state, and observation info into plain language. Then, it uses a Large Language Model to create the protocol. Also, it needs a clear link between numbers and their meanings. How well this works depends a lot on how well these descriptions are written.

---

> ### Author Rebuttal · Authors · 2026-03-31
>
> We sincerely thank the reviewer for the thoughtful and in-depth questions. We would like to address the reviewer’s concerns as follows. We will incorporate all of the points addressed in our response into the revised manuscript.
>
> ---
>
> **Weakness 1 \& 3, Question 1 (Dependence on input prompt design).**
>
> We appreciate the reviewer’s concern and agree that this is an important issue. Since several reviewers raised a similar question, in accordance with the rebuttal guidelines, we kindly refer the reviewer to our response to Reviewer gWcq (Table R3), where we provide additional experiments that simplify both $\mathcal{I}_T$ and $\mathcal{I}_P$. Specifically, for the task description $\mathcal{I}_T$, we consider a simplified version that provides only a basic state description and observation-chunk-level information, rather than a dimension-wise description. We also substantially simplify the communication design instruction $\mathcal{I}_P$ so that it includes only the communication objective and the output format. Based on this setup, we evaluate several variants, and the results in Table R2 show that **LMAC still operates successfully under these simplified prompts**, indicating that **its performance is not highly sensitive to the input prompt design**. While some minimal prompt specification may still be required for a new environment, we believe this remains well within a practical range.
>
> ---
>
> **Weakness 2 (Additional complexity)**
>
> We agree with the reviewer that LMAC introduces some additional complexity, since it uses an LLM to design the communication protocol. However, this overhead arises only during the design stage. During online execution, the method simply applies the designed protocol and incurs no extra complexity. Moreover, as analyzed in Appendix E.3, the computational cost of the underlying MARL algorithm is comparable to that of other communication-based methods. Since the use of an LLM is central to our goal of designing effective communication protocols, and the required token usage is also modest, we believe the overall approach remains practical.
>
> ---
>
> **Question 2 (Validity under small environment changes).**
>
> We agree with the reviewer that it is important to verify whether the designed communication protocol remains effective when the environment changes. Reviewer USMG raised a similar question, so we conduct additional experiments on bane_vs_hM by changing the agents’ spawn positions or further restricting their visibility. The protocol is learned in the original environment with fixed agent positions, and we then evaluate whether it remains effective under these modified settings. Table R2 reports the results. The results show that the protocol is not always optimal under the changed settings and that performance drops slightly, but it continues to work well overall and still provides meaningful gains. We believe this is because **the proposed LMAC protocol is not tied to fixed positions or visibility patterns**, but is designed to compensate for missing information under partial observability. As a result, it remains effective under moderate environment changes.
>
> **Table R2.** Comparison on bane_vs_hM under modified environment settings (Win rate).
>
> | LMAC (Original) | LMAC (Spawn switch) | LMAC (Visibility reduction) | QMIX |
> | --- | --- | --- | --- |
> | $\mathbf{75.2 \pm 3.5}$ | $73.4 \pm 3.9$ | $74.2 \pm 4.1$ | $19.5 \pm 2.1$ |
>
> ---
>
> **Question 3 (Effect of protocol constraints)**
>
> Regarding the protocol-related constraints, we provide the following qualitative observations. When the non-parametric constraint was removed, the LLM occasionally generated parameterized neural-network-based code, which not only requires additional optimization before use but also often fails to produce the expected output shape. As a result, the code regeneration process described in Appendix A.2 was triggered much more frequently. When the explicitness constraint was relaxed, we observed that the messages became heavily compressed. These observations suggest that such constraints do help improve the stability of protocol generation to some extent. However, they are not strictly necessary, and our experiments above show that communication for information sharing still works well even without these fine-grained prompt designs.
>
> ---
>
> We once again thank the reviewer for the thoughtful feedback, and we hope that our responses have adequately addressed the reviewer’s concerns.

---

> > ### Author Rebuttal · Reviewer_sokR · 2026-04-01
> >
> > Thank you for your latest response. Most of my concerns have been addressed by the rebuttal. I believe this multi-agent communication framework could be useful to the community, so l am inclined to support acceptance. I have therefore raised my score to 4 and am willing to discuss further with fellow reviewers.

---

> > > ### Author Response · Authors · 2026-04-04
> > >
> > > We sincerely thank the reviewer for the thoughtful feedback and for the improved evaluation of our work. We will incorporate the responses discussed in the review into the revised manuscript. We thank the reviewer again for their valuable comments.

---

### Official Review · Reviewer_1DmZ · 2026-03-13

**Soundness:** 2
**Presentation:** 3
**Significance:** 2
**Originality:** 3
**Overall Recommendation:** 3
**Confidence:** 4

**Summary:**

This paper proposes LMAC, which leverages an LLM to interatively refine the code-based communication protocol to help reconstruct the global state and mitigate the inter-agent information imbalance. Empirical studies demonstrate the effectiveness in state recovery and performance gains over communication baselines.

**Compliance With Llm Reviewing Policy:**

Affirmed.

**Final Justification:**

Thanks to the authors for the explanation. My major concern is the applicability of the proposed framework to high-dimensional observations, which largely remains after the rebuttal, as I do not see a clear and practical adaptation given by the authors. As a result, this represents a significant challenge for the current method.

Given the integrations of LLMs, I believe it is reasonable to ask for higher standard for the method, so I tend to keep the original score.

**Key Questions For Authors:**

Please refer to the weaknesses.

**Limitations:**

Yes

**Strengths And Weaknesses:**

Strengths:
The idea of designing executable communication protocols with an LLM to reconstruct the state is novel, and the empirical results are strong across several benchmarks.

Weaknesses:
1. Synthesizing communication messages through code implicitly assume a low-dimensional vector-based observation space, it is not clear how to apply this method to high-dimensional image-based observation space.
2. For this method to work, it requires natural-language descriptions for every state and observation dimension. This per-dimension semantic annotation is environment-specific and must be manually prepared for each new scenario, limiting its applicability.
3. Reconstructing the full global state may not be necessary for effective coordination, given that the LLM already has access to rich semantic information within the observations and task description.
4. The quality of the learned communication protocol may be highly dependent on the offline dataset collected in the beginning.

---

> ### Author Rebuttal · Authors · 2026-03-31
>
> We sincerely thank the reviewer for the thoughtful and in-depth questions. We would like to address the reviewer’s concerns as follows. We will incorporate all of the points addressed in our response into the revised manuscript.
>
> ---
>
> **Weakness 1 (Applicability to image-based observations).**
>
> We appreciate the reviewer’s thoughtful suggestion. While our paper focuses on vector-based observations, which are common in many MARL environments, we believe **the proposed approach can also be extended to image-based observations by combining it with a vision-language model (VLM).** Specifically, a VLM could first convert image observations into language descriptions, after which an LLM could reason over these descriptions and determine which information should be shared or masked. We believe this is a highly interesting direction for future work, and we thank the reviewer for this insightful comment.
>
> ---
>
> **Weakness 2 (Dependence on dimension-wise language descriptions).**
>
> We appreciate the reviewer’s concern and agree that this is an important issue. Since several reviewers raised a similar question, in accordance with the rebuttal guidelines, we kindly refer the reviewer to our response to Reviewer gWcq (Table R3), where we provide additional experiments that simplify both $\mathcal{I}_T$ and $\mathcal{I}_P$. Specifically, for the task description $\mathcal{I}_T$, we consider a simplified version that provides only a basic state description and observation-chunk-level information, rather than a dimension-wise description. We also substantially simplify the communication design instruction $\mathcal{I}_P$ so that it includes only the communication objective and the output format. Based on this setup, we evaluate several variants, and the results in Table R2 show that **LMAC still operates successfully under these simplified prompts**, indicating that **its performance is not highly sensitive to the input prompt design**. While some minimal prompt specification may still be required for a new environment, we believe this remains well within a practical range.
>
> ---
>
> **Weakness 3 (Necessity of full global-state reconstruction).**
>
> We agree with the reviewer’s point that effective coordination does not always require reconstructing the full global state. When the meaning of each state dimension is clearly specified, the system could instead be designed so that the LLM reconstructs only the task-relevant state information needed for coordination. However, when considering more practical settings, such as simplified prompts or extensions to image-based observations as suggested by the reviewer, the semantics of individual dimensions may no longer be clearly specified. **In such cases, reconstructing the full state can be a more practical design choice** than requiring the model to recover only selected dimensions.
>
> ---
>
> **Weakness 4 (Effect of dataset quality).**
>
> We thank the reviewer for raising this important concern. Although LMAC is trained using an offline dataset collected during training, we agree that its performance could be affected by the quality of the initial dataset. To examine this, we additionally evaluate LMAC using a dataset generated by a random policy, and report the results in Table R1. The results show that **LMAC is not highly sensitive to dataset quality**, as the performance gap remains small. We believe this is because LMAC is designed not to imitate an optimal policy, but to learn a communication strategy that reconstructs state information effectively across diverse situations. As a result, its performance depends less on the quality of the initial offline dataset.
>
> **Table R1.** Comparison of LMAC under different initial datasets (Win rate).
>
> | Dataset | 1o_10b_vs_1r | bane_vs_hM |
> | --- | --- | --- |
> | LMAC (random dataset) | $95.7 \pm 1.8$ | $74.3 \pm 3.6$ |
> | LMAC (training-collected dataset) | $\mathbf{96.2 \pm 1.6}$ | $\mathbf{75.2 \pm 3.5}$ |
>
> We once again thank the reviewer for the thoughtful feedback, and we hope that our responses have adequately addressed the reviewer’s concerns.

---

> > ### Author Rebuttal · Reviewer_1DmZ · 2026-04-03
> >
> > We thank the author for the clarification. However, some of the concerns remain unresolved:
> > 1. Extending the framework to image-based observations via a VLM is non-trivial, and I did not see a feasible way for this. Processing each observation through a VLM at every timestep would introduce substantial computational overhead, and it remains unclear how VLM-generated text descriptions would be compatible with the code-based communication protocol.
> > 2. As acknowledged, reconstructing the full state can be a more practical design choice when the semantics of individual dimensions are not clearly specified. However, the code-based protocol fundamentally depends on per-dimension semantics to generate hardcoded indexing operations. If such semantics are absent, the LLM cannot produce the protocol in the first place, leaving the method inapplicable in the settings where full reconstruction is claimed to be most useful.

---

> > > ### Author Response · Authors · 2026-04-04
> > >
> > > Thank you for the reviewer’s follow-up questions. We will provide a more detailed response to both questions.
> > >
> > > **1. Regarding the use of VLMs with image observations.** We agree with the reviewer that converting all real-time images through a VLM would be prohibitively expensive. Instead, as suggested in the main paper, one possible approach is to use offline data to learn the relationship between image observations and states, and then reason about which semantic information should be communicated and how it can be structured. Based on this, one could train a separate CNN-based decoder that maps image observations into semantic information, and use such information for communication.
> > >
> > > Specifically, consider the SMAC example. Suppose that, through offline VLM-based analysis, we identify that **the red-colored agents correspond to enemies and that their positions are the key information** needed for task completion. Then rather than using a VLM we could train a separate CNN decoder, for example using object detection, to infer the pixel positions of red agents from the image. **These positions (=semantic info.) can then be shared among agents from the proposed communication protocol.** In this way converting images into semantic representations makes it possible to selectively transmit only relevant information without requiring real-time VLM inference.
> > >
> > > While this example may seem complex, selective communication under image observations is inherently challenging. We believe this is an important future direction and thank the reviewer for highlighting this point.
> > >
> > > **2. Regarding the need for semantic descriptions.** We would like to clarify a possible misunderstanding: Our claim was that full-state reconstruction can still be effective **when semantic information is unavailable for the state, not when semantic information is unavailable for the observation.** When semantic information about the state is unavailable, our implementation is still entirely feasible, as shown by **LMAC (V2) in Table R3**, with no substantial performance difference. However, when the observation contains no semantic information at all, an additional process may be needed to construct such semantics, as in the image-based example discussed above.
> > >
> > > In summary, although our current framework assumes some degree of semantic information in the observations, it can be extended beyond this setting. We believe that a key contribution of this work is to demonstrate the potential role and impact of language communication in MARL. We thank the reviewer for the valuable feedback.

---

### Decision · Program_Chairs · 2026-04-30

**Decision:**

Accept (regular)

**Comment:**

The paper has strong strengths in novel LLM-guided communication design and strong empirical results across benchmarks; three reviewers were happy that their concerns were fully resolved.

While Rev 1DmZ’s concern about high-dimensional image inputs is reasonable, it is not a requirement for every MARL paper – especially one that explicitly focuses on vector-based observations.

Three reviewers recommend Weak Accept, one recommends weak Reject. The disagreement centers on whether the reliance on semantic descriptions and the unclear path to high-dimensional observations are fatal flaws. Given that the method targets vector-observation MARL benchmarks (which are standard in the field) and the authors demonstrate substantial gains with practical prompt engineering, the majority view is that the paper offers a valuable contribution that others are likely to build upon.

I recommend accepting this paper.